

# Effect of legume intercropping on N₂O emission and CH₄ uptake during maize production in the Ethiopian Rift Valley

Shimelis G Raji[1,2] and Peter Dörsch[1]

[1] Faculty for Environmental Sciences and Resource Management, Norwegian University of Life Sciences (NMBU), 1432 Ås, Norway

[2] College of Agriculture, Hawassa University, P.O, Box 05, Hawassa, Ethiopia

Correspondence to: Peter Dörsch (peter.doersch@nmbu.no)

## Abstract

Intercropping with legumes is an important component of climate smart agriculture (CSA) in sub Saharan Africa, but little is known about its effect on soil greenhouse gas (GHG) exchange. A field experiment was established at Hawassa in the Ethiopian rift valley, comparing nitrous oxide (N₂O)

and methane (CH₄) fluxes in minerally fertilized maize (64 kg N ha$^{-1}$) with and without crotalaria (*C. juncea*) or lablab (*L. purpureus*) as intercrops over two growing seasons. To study the effect of intercropping time, intercrops were sown either three or six weeks after maize. The legumes were harvested at flowering and half of the above-ground biomass was mulched. In the first season, cumulative N₂O emissions were largest in 3-week lablab, with all other treatments being equal or

lower than the fertilized maize monocrop. After reducing mineral N input to intercropped systems by 50% in the second season, N₂O emissions were at par with the fully fertilized control. Maize yield-scaled N₂O emissions in the first season increased linearly with above-ground legume N-yield (p=0.01), but not in the second season when early rains resulted in less legume biomass because of shading by maize. Growing season N₂O-N emission factors varied from 0.02 to 0.25

and 0.11 to 0.20% of the estimated total N input in 2015 and 2016, respectively. Growing season CH₄ uptake ranged from 1.0 to 1.5 kg CH₄-C ha$^{-1}$ with no significant differences between treatments or years, but setting off the N₂O-associated global warming potential by up to 69%. Our results suggest that high yielding leguminous intercrops entail some risk for increased N₂O emissions when used together with recommended fertilization rates, but can replace part of the





fertilizer N without compromising maize yields in the following year and thus support CSA goals while intensifying crop production in the region.

Key words: yield-scaled $N_2O$ emissions, $CH_4$ uptake, legume-intercropping, maize, Africa

## 1. Introduction

With a rapidly increasing population and declining agricultural land in Sub-Saharan Africa (SSA), increasing productivity per area (intensification) is the only viable alternative for producing sufficient food and feed (Hickman et al., 2014a). Intensification entails increased use of inorganic fertilizers, which may cause $N_2O$ emissions and reduce the soil $CH_4$ sink (Castro et al., 1994, Xie

et al., 2010). Climate smart agriculture (CSA), by contrast, has been proposed as a way forward to simultaneously increase agricultural productivity and profits, while increasing climate resilience and reducing greenhouse gas (GHG) emissions (Neufeldt et al., 2013). However, understanding of greenhouse gas emissions from crop production in SSA in general and CSA in particular is limited and the potential of crop production in SSA as a source or sink of the greenhouse gases $CO_2$, $N_2O$,

and $CH_4$ is understudied (Kim et al., 2016, Hickman et al., 2014b). Moreover, modelling studies predict significant negative impacts of climate change on crop productivity in Africa (Blanc and Strobl, 2013) and it is largely unknown how these and the countermeasures taken to maintain agricultural productivity will affect GHG emissions.

Crop production is a major source of nitrous oxide ($N_2O$), the third-most important anthropogenic

GHG after $CH_4$ and $CO_2$ (IPCC, 2014). Inorganic and organic N added to soil provide ammonium ($NH_4^+$) and nitrate ($NO_3^-$) for nitrification and denitrification, respectively, which are the two main processes of microbial $N_2O$ production in soil (Khalil et al., 2004). The rate of $N_2O$ formation in upland soils depends greatly on the extent and distribution of anoxic microsites, which is controlled by soil moisture, texture and the distribution of decomposable organic matter and $NH_4^+$ fueling

heterotrophic and autotrophic respiration, respectively (Schlüter et al., 2019, Wrage-Mönnig et al., 2018). The magnitude of soil $N_2O$ emissions depends on $O_2$ availability as controlled by soil moisture and respiration, availability of mineral N and readily decomposable C (Harrison-Kirk et al., 2013) and soil pH (Russenes et al., 2016), all of which depend on soil management practices. The $N_2O$ yield of nitrification (Nadeem et al., 2019) and the production and reduction of $N_2O$



during denitrification (Bakken et al., 2012) are further controlled by soil pH and by the balance
      between oxidizable carbon and available $NO_3^-$ (Wu et al., 2018). Mulching and incorporation of
      crop residues leads to increased N mineralization and respiratory $O_2$ consumption, thus potentially
      enhancing $N_2O$ emissions both from nitrification and denitrification (Drury et al., 1991), if soil
      moisture is sufficient to support microbial activity and restrict $O_2$ diffusion into the soil.
Accordingly, $N_2O$ emissions are variable in time, often following rainfall events (Schwenke et al.,
      2016).

      Crop diversification by combining legumes with cereals, both in rotation and intercropping,
      enhances overall productivity and resource use efficiency (Ehrmann and Ritz, 2014). Intercropping
      of maize with grain legumes is common in the rift valley of Ethiopia and central in CSA (Arslan
et al., 2015). In low input systems common to the Rift Valley, integration of legumes with cereals
      diversifies the produce and improves the nitrogen nutrition of the cereal. Moreover, by partially
      replacing energy-intensive synthetic N, intercropping with legumes may increase the sustainability
      of the agroecosystem as a whole (Carranca et al., 2015). However, to make best use of the resource
      use complementarity of inter and main crop, the planting time of the intercrop has to be optimized
so that the maximum nutrient demand of the two components occurs at different times (Carruthers
      et al., 2000). The timing of intercrops could also affect $N_2O$ emissions if N mineralization from
      legume residues is poorly synchronized with the N requirement of the cereal crop. This can be
      counteracted by reducing mineral N additions to intercropping systems, but the timing of the
      intercrop (sowing date relative to the cereal crop) remains an issue that has, to the best of our
knowledge, not been studied with regard to $N_2O$ emissions.

      Intercropping and mulching may also affect the soil's capacity to oxidize atmospheric $CH_4$ as
      abundant $NH_4^+$ inhibits methanotrophs (Laanbroek and Bodelier, 2004). However, field studies
      with incorporation of leguminous or non-leguminous catch crops have been inconclusive (e.g.
      Sanz-Cobena et al., 2014). In a meta-study on $CH_4$ fluxes in non-wetland soils, Aronson and
Helliker (2010) concluded that N inhibition of $CH_4$ uptake is unlikely at fertilization rates below
      kg N $ha^{-1}$ $y^{-1}$ and that much to the contrary, N addition may stimulate $CH_4$ uptake in N-limited
      soils. Ho et al. (2015) found that incorporation of organic residues stimulated $CH_4$ uptake even in
      fairly N-rich Dutch soils. Intercrops may indirectly affect $CH_4$ uptake by lowering soil moisture
      and thus increase the diffusive flux of atmospheric $CH_4$ into the soil. Accordingly, Wanyama et





al. (2019) found $CH_4$ uptake to be negatively correlated with mean annual water-filled pore space in a study on different land use intensities in Kenya.

In a review on $N_2O$ fluxes in agricultural legume crops, Rochette and Janzen (2005) concluded that the effect of legumes on $N_2O$ emission is to be attributed to release of extra N by root exudation and decomposition of nodules rather than to the process of nitrogen fixation itself. Intercropped legumes may thus affect $N_2O$ emissions in two ways: by directly providing organic N or by modulating the competition between plants and microbes for soil N. Compared to mineral fertilizers, N supply from biological fixation is considered environmentally friendly as it can replace industrially fixed N (Jensen and Hauggaard-Nielsen, 2003), provided that crop yields remain the same. However, combining easily degradable crop residues with synthetic N can lead to elevated $N_2O$ emissions (Baggs et al., 2000), potentially compromising the environmental friendliness of intercropping in CSA. It is well known that the effect of crop residues on $N_2O$ emission depends on a variety of factors such as residue amount and quality (C:N ratio, lignin and cellulose content), soil properties (e.g. texture), placement mode (mulching, incorporation) and soil moisture and temperature regimes (Sanz-Cobena et al., 2014, Li et al., 2016). So far, there is only a limited number of studies addressing the effect of legume intercropping on $N_2O$ emissions and $CH_4$ uptake in SSA crop production (Baggs et al., 2000; Millar et al., 2004; Dick et al., 2008).

The main objective of the present study was to evaluate the effects of forage legume intercropping of maize on $N_2O$ and $CH_4$ emissions during maize production in the Ethiopian Rift Valley. We hypothesized that forage legumes increase $N_2O$ emissions and decrease $CH_4$ uptake depending on above-ground biomass, legume species and sowing date; legumes intercropped three weeks after sowing of maize would result in higher yields than those intercropped six weeks after maize and lead to increased $N_2O$ emissions if used with full-dose mineral fertilization. With late intercropping, legumes yields would be suppressed having no or little effect on $N_2O$ emission. Choosing legume species and sowing date and accounting for N inputs from legume intercrops, thus could allow to manage legume intercropping in SSA with reduced GHG emissions.

## 2. Materials and methods


## 2.1 Study area

The field experiment was conducted at the Hawassa University Research Farm, 07°3'3.4"N and 38°30"20.4'E at an altitude of 1660 m a.s.l. The mean annual rainfall is 961 mm, with a bimodal pattern. The rainy season between June and October accounts for close to 80% of the annual rainfall. Average maximum and minimum monthly temperatures are 27.4 and 12.9°C, respectively. The soil is a clay loam (46% sand, 26% silt, 28% clay), with a bulk density of 1.25 g cm$^{-3}$, a total

N content of 0.12%, an organic C content of 1.64 %, an available Olsen P content of 175 mg kg$^{-1}$ and a pH$_{H2O}$ of 6.14.

## 2.2 Experimental design and treatments

Experimental plots (20 m$^2$) with six treatments were laid out in a complete randomized block design (RCBD) with four replicates (Tab. 1). Seed bed was prepared in both years by mold board

plow to a depth of 0.25 m followed by harrowing by tractor. A hybrid maize variety, BH-540 (released in 1995) was sown on May 30 and May 7 in 2015 and 2016, respectively. Maize was planted at a density of 53,333 plants ha$^{-1}$. Following national fertilization recommendations, diammonium phosphate (18 kg N, 20 kg P) was applied manually at planting and urea (46 kg N) four weeks after sowing maize, except for the unfertilized control. The N fertilization rate was

halved for the intercropping treatments in the 2016 season to account for carry-over of N from forage legumes grown in the previous year. The forage legumes crotalaria (*C. juncea*) and lablab (*L. purpureus*) were planted between maize rows at a density of 500,000 and 250,000 plants ha$^{-1}$, respectively.

The above-ground forage legume biomass was harvested at flowering and half of it was removed.

The remaining half was spread manually between the maize rows after cutting the fresh biomass into ~10 cm pieces. As the mulching was done plot wise, plots within the same treatment received different amounts of mulch depending on the legume yield of each plot. In the 2016 growing season, all treatments were kept on the same plots as in 2015, capitalizing on plot-specific N and C input from previous mulch. Aboveground dry matter yield was determined by drying a

subsample at 72°C for 48 hours and C and N contents were measured by an element analyser.

## 2.3 N$_2$O and CH$_4$ fluxes and ancillary data

GHG exchange was monitored between the maize rows by static chambers (Rochette et al., 2008), using custom-made aluminum chambers with an internal volume of 0.144 m$^3$ and a cross-sectional



area of 0.36 m$^2$. Upon deployment, the chambers were pushed gently into the soil and sealed
around their circumference with moist clay to minimize leakage.

Sampling was carried out weekly during the period June to September, in 2015 and May to
September, in 2016 on 15 and 17 sampling dates, respectively. Gas samples were collected
between 9:00 AM and 2:00 PM. For each flux estimate, four gas samples were drawn from the
chamber headspace at 15 min intervals, using a 20 ml polypropylene syringe equipped with a 3-
way valve. Before transferring the sample to a pre-evacuated 10 cc serum vial crimp-sealed with
butyl septa, the sample was pumped 5 times in and out of the chamber to obtain a representative
sample. Overpressure was maintained to protect the sample from atmospheric contamination
during storage and shipment to the Norwegian University of Life Sciences, where the samples
were analyzed by gas chromatography. He-filled blank vials were included to evaluate
contamination, which was found to be less than 3% of ambient.

All samples were analyzed on a GC (Model 7890A, Agilent Santa Clara, CA, USA) connected to
an auto-sampler (GC-Pal, CTC, Switzerland). Upon piercing the septum with a hypodermic
needle, ca. 1 ml sample is transported via a peristaltic pump (Gilson minipuls 3, Middleton, W1,
USA) to the GC's injection system, before reverting the pump to backflush the injection system.
The GC is configured with two back-flushed pre-columns and a Poraplot U wide-bore capillary
column connected to a thermal conductivity, a flame ionization and an electron capture detector to
analyze $CO_2$, $CH_4$ and $N_2O$, respectively. Helium 5.0 was used as carrier and $Ar/CH_4$ (90:10
vol/vol) as makeup gas for the ECD. For calibration, two certified gas mixtures of $CO_2$, $N_2O$ and
$CH_4$ in He 5.0 (Linde-AGA, Oslo, Norway), one at ambient concentrations and one ca. 3 times
above ambient were used. A running standard (every tenth sample) was used to evaluate drift of
the ECD signal. Emission ($CO_2$, $N_2O$) and uptake ($CH_4$) rates were estimated by fitting linear ($R^2$
$\geq 0.85$) or quadratic functions to the observed concentration change in the chamber headspace and
converting them to area flux according to eq. 1

$$F_{GHG\,(\mu g\,m^{-2}h^{-1})} = \frac{dc}{dt} * \frac{Vc}{A} * \frac{Mn}{Vn} * 60 \qquad\qquad \text{Eq. (1)}$$

where, $F_{GHG}$ is the flux ($\mu$g $N_2$O-N m$^{-2}$ h$^{-1}$ in case of $N_2O$; $\mu$g $CH_4$-C in the case of $CH_4$), $\frac{dc}{dt}$ the
rate of change in concentration over time (ppm min$^{-1}$), $V_c$ the volume of the chamber (m$^3$), $A$ the
area covered by the chamber (m$^2$), $M_n$ the molar mass of the element in question (g mol$^{-1}$) and $Vn$





the molecular volume of gas at chamber temperature ($m^3$ $mol^{-1}$). A quadratic fit was only used in cases where $N_2O$ accumulation in the chamber showed a convex downwards and $CH_4$ uptake a convex upwards trend (i.e. decreasing emission or uptake rates with time) to estimate time-zero rates. Fluxes were cumulated plot-wise by linear interpolation for each growing season.

In 2016, soil moisture and temperature at 5 cm depth were monitored hourly using data loggers (Decagon EM50, Pullman, WA, USA) together with $ECH_2O$ sensors (Decagon) for volumetric soil water content (VSWC) and temperature at five points across the experimental field. The sensors were placed in control, M+Cr3w and M+Lb3w (2). No data are available for the 2015 season, due to equipment failure.

Intact soil bulk density and an assumed particle density of 2.65g $cm^{-3}$ were used to calculate daily water filled pore space values for the 2016 growing season:

$$WFPS = VSWC/(1 - \frac{BD}{PD}) * 100 \qquad \text{Eq. (2)}$$

where *WFPS* is the water filled pore space, *VSWC* the volumetric soil water content, *BD* the bulk density and *PD* the particle density which was set to 2.65 g $cm^{-3}$. Daily rainfall data were collected using an on-site rain gauge monitored daily during the growing season.

**2.4 Estimating N inputs and $N_2O$ emission factors**

N input from forage legume crop residues was estimated from measured above-ground dry matter yield, its N content and the amount of mulch applied. To account for belowground inputs a shoot to root ratio of two was assumed for both crotalaria and lablab (Fageria et al., 2014). Dry matter yields of forage legumes differed greatly depending on sowing time, with generally larger yields in 3-week than 6-week intercropping. Also, forage legumes sown three weeks after maize grew faster and were harvested and mulched earlier than those sown six weeks after maize. We assumed that 50% of the legume N (mulched and belowground) was released during the growing season but reduced this amount to 30% for the aboveground component (mulch) of the 6-week treatments to account for the later mulching date. The proportions becoming available during the growing seasons are conservative estimates based on Odhiambo (2010), who reported that about 50% of N contained in crotalaria, lablab and mucuna was released during a 16-week incubation experiment at optimal temperature and moisture conditions. Placing litter bags into dry surface soil, Abera et





al. (2014) found that legume residues decomposed rapidly under *in situ* conditions in the Ethiopian Rift Valley, releasing up to 89% of the added N within 6 months.

For the second year, 50% of the N left after the growing season (below and aboveground) was assumed to become available, on top of the N-input from the newly sown forage legumes. Dry matter yields of forage legumes and estimated N input for the two years are presented in table 1.

Treatment-specific, growing-season $N_2O$ emission factors were calculated as:

$$N_2O\ EF = \frac{(N_2O_{treatment} - N_2O_{control})}{N\ input} * 100 \qquad \text{Eq. (3)}$$

where $N_2O\ EF$ is the $N_2O$ emission factor (% of N input lost as $N_2O$-N), $N_2O_{treatment}$ the cumulative $N_2O$-N emission (from sowing to harvest) in the fertilized and intercropped treatments, $N_2O_{control}$ the emission from the 0N0P treatment (background emission) and $N_{input}$ the estimated total input of N.

## 2.5 Grain yields and yield-scaled N₂O emissions

Maize grain yield was determined by manually harvesting the three middle rows (to avoid border effects) of each plot, and was standardized to 12.5% moisture content. All values were extrapolated from the plot to the hectare. To estimate yield-scaled $N_2O$ emissions (g $N_2O$-N ton$^{-1}$ grain yield), cumulative emissions were divided by grain yield.

## 2.6 Statistical analysis

Differences in cumulative $CH_4$ and $N_2O$ emissions between treatments in each cropping season were tested by analysis of variance (ANOVA) with LSD used for mean separation after testing the data for normality and homoscedasticity. Cumulative seasonal $N_2O$ emissions for 2015 were log-transformed. Statistical significance was declared at $P \leq 0.05$.





# 3. Results

## 3.1 Weather conditions

The year 2015 was one of the most severe drought years in decades and, as a result, sowing in 2015 was delayed by 3 weeks as compared to 2016. Rain fell late during the growing season and the cumulative rainfall for April to October was about 100 mm lower in 2015 than in 2016 (Fig. 1d and 1g).

## 3.2 $N_2O$ fluxes

$N_2O$ emission rates in 2015 (treatment means, n=4) ranged from 1.1 to 13.7 µg N m$^{-2}$ h$^{-1}$ for the control treatment, with no obvious peaks (Fig. 1a). Similarly, for fertilized maize, $N_2O$ emissions ranged from 2 to 23.5 µg N m$^{-2}$ h$^{-1}$. Emission fluxes were generally larger for the intercropped treatments: crotalaria treatments emitted $N_2O$ at rates of 1.7 - 34.3 and 2.1 – 24.2 µg N m$^{-2}$ h$^{-1}$ when intercropped 3 or 6 weeks after maize, respectively, while maize-lablab emitted 1.9 – 62.7 µg N m$^{-2}$ h$^{-1}$ when sown 3 weeks and 1.5 - 10.7 µg N m$^{-2}$ h$^{-1}$ when sown 6 weeks after maize. The generally low emission rates in the latter system (6-weak lablab intercropping) corresponded to poor growth of lablab due to shading by the maize plants. Irrespective of legume species, highest emission rates were found for intercrops planted three weeks after maize (Fig. 1b and 1c). A peak of $N_2O$ emission occurred in the 3-week maize-lablab system around mid-August, 2015, which was significantly larger than in the control (P=0.013), fertilized maize monocrop (P=0.001), or crotalaria (P=0.021) and lablab (P=0.002) intercropped 6 weeks after maize.

During the 2016 season, $N_2O$ emission rates in the 0N-control varied between 2.5 and 22.8 µg N m$^{-2}$ h$^{-1}$, peaking at the beginning of the season when WFPS was >50%. There were no significant differences in WFPS values between treatments (data not shown). Fertilized maize had similar rates (3.1 - 24.2 µg N m$^{-2}$ h$^{-1}$) peaking at around four weeks after planting. Maize-forage legume treatments had larger emission rates, ranging from 1.8 to 40.2 and 3.2 to 58.6 for crotalaria planted 3 and 6 weeks after maize, respectively and 3.9 to 38.0 and 1.9 to 45.2 µg N m$^{-2}$ h$^{-1}$ for lablab planted 3 and 6 weeks after maize, respectively. In general, emission rates were higher in the beginning than in the end of the cropping season (Fig. 1d-f). Despite higher fluxes for intercropping treatments than in the unfertilized control in week 1 (P=0.162) and 4 (P=0.061), there were no statistically significant differences in flux rates between the treatments.



### 3.3 Cumulative N$_2$O emissions

During the 2015 growing season, all treatments had equal or higher cumulative N$_2$O emissions
than the unfertilized control, with the 3-week lablab intercropping system emitting significantly
more N$_2$O than the unfertilized control (p=0.006) and the 6-week lablab intercrop (Fig. 2a).
Comparing intercropping treatments with the fertilized control, lablab sown three weeks after
maize clearly increased N$_2$O emissions but not significantly (P=0.35), whereas all other
intercropping treatments had cumulative N$_2$O emissions comparable with fertilized maize control.
Regarding sowing date, 3-week lablab had significantly higher N$_2$O emissions (P<0.01) than its 6-
week counterpart, whereas no such effect was seen for crotalaria.

During the 2016 growing season, lablab intercropping 3-weeks after maize showed significantly
higher (P<0.01) cumulative N$_2$O emissions than the unfertilized control, but there was no
difference between fully fertilized maize monocrop and intercropped maize treatments fertilized
with 50% of the mineral N applied in 2015, nor was there any effect of intercropping date (3 vs. 6
weeks; Fig. 2b).

### 3.4 Legume and maize yields

Aboveground yields of lablab were generally higher than those of crotalaria (Table 1).
Intercropping three weeks after maize resulted in higher biomass yields compared to six weeks for
both legume species. Both legumes grew poorly during the second growing season, particularly
crotalaria. Maize grain yields differed greatly between the years and were roughly 20% higher in
the wetter year of 2016 (Table 2). Better growth conditions for maize in the second year resulted
in smaller yields of intercrop legumes.

### 3.5 N$_2$O emission factor and intensity

Growing-season emission factors (EF) varied from 0.02 to 0.25 and 0.11 to 0.20% in 2015 and
2016, respectively (Table 2). Of the intercropped treatments, lablab intercropped three weeks after
maize resulted in a significantly larger emission factor than fertilized maize and other
intercropping treatments, whereas there was no significant difference in 2016. Overall, growing-
season N$_2$O emission factors were ~ 40% higher in 2016 than in 2015, which is mainly due to the
smaller N input in 2016 which was 25 to 45% lower than in 2015, except for the 3-week lablab
system which had an estimated 18% higher N input in 2016 than 2015 (Table 1). The latter was



due to the extraordinary high lablab yield in the previous year and its stipulated carryover (Table 1).

Mean yield-scaled $N_2O$ emissions in 2015 varied between 25 to 55 g $N_2O$ ton$^{-1}$ grain yield. In
2015, 3-week lablab had a higher $N_2O$ intensity than 6-week lablab, whereas all other differences were insignificant. In 2016, with mineral N fertilization reduced to 50%, $N_2O$ emission intensities varied from 26 to 37 g $N_2O$ ton$^{-1}$ grain, with no significant effect of legume species, sowing date or N fertilization (Table 2).

To further explore the variability of $N_2O$ emissions, we plotted cumulative $N_2O$ emissions plot-
wise against legume N yield, but found no relationship (not shown). However, when plotting yield-scaled $N_2O$ emission over legume N yield, a significant positive relationship (P=0.01) emerged for 2015, but not 2016 (Fig. 3a and 3b), suggesting that leguminous N input increased $N_2O$ emissions more than maize yields in the dry year of 2015.

### 3.6 $CH_4$ fluxes

All treatments acted as net sink for $CH_4$, with uptake rates ranging from 31 to 93 µg C m$^{-2}$ h$^{-1}$ in 2015 (Fig. 4a-c). Uptake rates in 2015 were rather constant in time with somewhat elevated uptake rates towards the end of the season. There were no obvious treatment effects. By contrast, in the wetter year of 2016, $CH_4$ uptake showed a pronounced maximum in the beginning of June with uptake rates of up to 140 µg C m$^{-1}$ h$^{-1}$ irrespective of treatment (Fig. 4d-f), when WFPS values
declined to values below 25% (Fig. 4g). Methane uptake during this period tended to be greatest in the unfertilized control, while intercropping treatments had smaller uptake rates, which, however, were not significantly different from maize monocrop treatments. Differences between treatments at single sampling dates were insignificant throughout the season. Highest $CH_4$ uptake in 2016 was recorded with lowest WFPS (~10%).

### 3.7 Cumulative $CH_4$ uptake

Cropping season cumulative $CH_4$ uptake exceeded 1 kg C ha$^{-1}$ in both years with no significant effect of intercropping, legume species or time of intercropping (Fig. S1a and S1b). Plots intercropped with crotalaria tended to take up less $CH_4$ but this effect was not statistically significant in neither 2015 nor 2016 (P=0.056). Plotting cumulative $CH_4$ uptake plot-wise over
legume dry matter yield did not result in a significant relationship, but highest seasonal uptake rates occurred in plots with lowest legume dry matter yield (data not shown).





### 3.8 Non-CO$_2$ GWP

Non-CO$_2$ global warming potentials (GWP) were calculated as CO$_2$ equivalents balancing cumulative seasonal N$_2$O-N emissions with CH$_4$ uptake on the plot level and averaging them for

treatments (Table 2, Fig. 5). The relative contribution of CH$_4$ to the non-CO$_2$ GWP of the different cropping systems varied between 22 and 69% and was highest in the non-fertilized maize monocrop. Three-week lablab intercropping resulted in significantly higher GWP compared with 6-week lablab intercropping and maize mono-cropping (Table 2). By contrast, in 2016, legume species but not intercropping time affected the GWP balance (P<0.05). Lablab intercropped 3

weeks after maize resulted in significantly higher (P<0.05) GWP than the unfertilized control but was indistinctive from the fertilized maize monocrop, or other intercrop treatments (Table 2, Fig. 5a and 5b).

## 4. Discussion

### 4.1 Maize-legume intercropping and N$_2$O emissions

Background N$_2$O emissions (in unfertilized maize monocrop) fluctuated between 1.1 and 23 µg N$_2$O-N m$^{-2}$ h$^{-1}$, which is in the range of previously reported emission rates for soils in SSA with low N fertilizer input (Pelster et al., 2017). Baseline emissions were somewhat higher in the wetter season of 2016, owing ~100 mm more rainfall (Fig. 1d and 1g). Elevated emission rates >30 µg

N$_2$O-N m$^{-2}$ h$^{-1}$ occurred in 2015 on few occasions in intercrop treatments, notably in mid-August when rainfall occurred right after mulching of the three-week intercrops. Mulching of the six-week intercrops did not affect N$_2$O emission, probably because the mulched legume biomass too small to affect the flux (Fig. 1b, 1c; Table 1). In 2016, mulching of the 3-week legumes was followed by rainfall, increasing the WFPS to 50% (Fig. 1g), however, without resulting in elevated N$_2$O

emission rates (Fig. 1e, 1f). Together, this suggests that the direct effect of mulching on N$_2$O emission depends on soil moisture and the amount of mulched biomass, and can hence not be generalized.

Legume dry matter yields varied strongly (100 to 3000 kg ha$^{-1}$) throughout the two experimental years (Table 1, Fig. 3), depending on species, intercropping time and weather. Three-week

intercrops performed generally better than six-week intercrops, which appeared to be inhibited in



growth by shading through maize. This was particularly apparent for the low-growing lablab legume. In terms of legume biomass, lablab grew more vigorously and realized larger dry matter yields than crotalaria (Table 1). Moreover, lablab is known to be a better $N_2$ fixer than crotalaria (Ojiem et al., 2007). Together, this resulted in a wide range of potential leguminous N-inputs in

our experiment, which could be used to examine their overall effect on $N_2O$ emission under Ethiopian rift valley conditions on a seasonal basis. Surprisingly, we did not find any significant relationship between estimated total N input or legume N yield and cumulative $N_2O$ emission. This may be due to the notoriously high spatial and temporal variability of $N_2O$ emissions rates within treatments, or reflect the fact that intercropping had no or opposing effects on $N_2O$ forming

processes. Cumulative $N_2O$ emissions and legume N yields integrate over the entire season and do not capture seasonal dynamics of soil N cycling and N uptake, which could obscure or cancel out transient legume effects on $N_2O$ emissions. Possibly, N released in intercropping treatments was effectively absorbed by the main crop, even though intercropping did not lead to significantly higher maize grain yields in our experiment. Alternatively, changes in physicochemical conditions

brought about by intercrops, such as potentially lower soil moisture due to more evapotranspiration, may have counteracted the commonly observed stimulating effect of legume N on $N_2O$ emissions (Almaraz et al., 2009, Sant'Anna et al., 2018).

To further elucidate the $N_2O$ emission response to legume intercropping, we plotted cumulative $N_2O$ emissions normalized for grain yields ("$N_2O$ intensity") plot-wise over measured legume N

yields, thereby utilizing the wide range of potential leguminous N inputs provided by our experiment. A significantly positive relationship between $N_2O$ intensity and legume N yields emerged for 2015, suggesting that intercropped legumes indeed increase $N_2O$ emissions relative to maize yields (Fig. 3a). It is impossible to say, however, whether this relationship was driven by the extra N entering the system through biological N fixation, or whether an increasing legume

biomass affected physicochemical conditions in the rhizosphere favoring $N_2O$ formation. In 2016, legume dry matter yields were much lower than in 2015, owing early rains favoring maize growth, and no significant relationship with $N_2O$ intensity was found (Fig. 3b). This illustrates that the effect of legume intercropping on $N_2O$ emissions is highly dependent on sowing date and weather, both of which control the growth of legume and main crop and ultimately the amount and fate of

leguminous N in the intercropping system. Our data suggest that excessive accumulation of leguminous biomass in SAA maize cropping enhances the risk for elevated $N_2O$ emissions.





We expected $N_2O$ emissions to respond more strongly to intercropping in the second year (2016), as legume mulches were applied according to their plot-wise aboveground yields in the previous year. Indeed, $N_2O$ emission rates were clearly higher in intercropping plots on the first sampling date in 2016 (fig. 1e and 1f), indicating increased N cycling in mulched plots. This difference vanished quickly, however, suggesting that the effect of intercrop mulches, even at high amounts (Table 1), on $N_2O$ emissions in the subsequent year is negligible under SSA conditions. It is noteworthy that our estimates of the fraction of N carried over between the years were based on literature data (Table 1), and that a considerable part of the mulched N may have been lost during abundant rainfalls (300 mm) early in the 2016 season before crops were sown.

It is striking that cumulative $N_2O$ emissions were at par with the fully fertilized maize monocrop in 2016. This effect, however, was short-lived and no significant difference in average flux rates was seen during the remainder of the season resulting in statistically indistinguishable cumulative $N_2O$ emissions. This may be partly due to the 50% reduction in mineral N application to intercrop treatments, as found by others (Tang et al., 2017). Another reason may be that a considerable proportion of the cumulative emission in 2016 occurred before or shortly after 3-week intercrops were sown, and was thus unaffected by growing legumes. Overall, cumulative $N_2O$ emissions were equal or higher in 2016 than in 2015, despite reduced mineral N addition to intercrops and lower legume biomass. Ultimately, the lack of a clear emission response to legume intercropping in the second year calls for studies tracing cumulative mulching effects over multiple years. In our study, amount and timing of rainfall appeared to be more important for $N_2O$ emissions in the second year than amount and carryover of legume N.

Given our finding that $N_2O$ intensity responded positively to legume biomass and its N content in a drought year with poor maize growth, intercrop species and sowing and harvest date (relative to the main crop) emerge as viable management factors for controlling $N_2O$ emissions in SSA intercropping systems. Legume species and cultivar in intercropping systems are known to be critical for N loss, both during the intercropping and the subsequent seasons (Pappa et al., 2011, Weiler et al., 2018). The stimulating effect of crop residues on $N_2O$ emission has been reported to depend on residue quality and soil moisture, with denitrification being the likely process (Li et al., 2016). Our study provides evidence that vigorous growth of high yielding legume intercrops can enhance $N_2O$ emissions in years unfavorable for maize growth, whereas in years with sufficient water availability early in the growing season, maize growth is favored preventing excessive





growth of the intercrop. Our study therefore points to sowing date as the most promising option to control growth of the intercrop relative to the main crop and hence to deal with the risk of increased

$N_2O$ emissions with legume intercrops.

## 4.2 Seasonal $N_2O$ and $CH_4$ emission, $EF_{N2O}$ and GWP

Growing season $N_2O$ emissions in fertilized treatments varied from 0.17 to 0.33 and 0.23 to 0.3 kg $N_2O$-N ha$^{-1}$ in 2015 and 2016 covering 107 and 123 days, respectively (Fig. 2), and a range of total N inputs from 36.4 to 97.8 kg N ha$^{-1}$ (Table 1). There are no $N_2O$ emissions studies for maize-

legume intercropping in the Ethiopian Rift valley so far. Hickman et al. (2014a) reported $N_2O$ emissions of 0.62 and 0.81 kg N per ha and 99 days for 100 and 200 kg N input ha$^{-1}$, respectively, for a maize field without intercropping in humid western Kenya. Baggs et al. (2006), working in the same region with maize intercropped with legumes in an agroforestry system reported $N_2O$ emissions ranging from 0.2 to 0.6 kg N ha$^{-1}$ with higher emissions in tilled intercropping

treatments. The largest seasonal $N_2O$ emission for intercropping reported so far from SSA is 4.1 kg N ha$^{-1}$ (84 days) after incorporating 7.4 t ha$^{-1}$ of a *Sesbania-Macroptilium* mixture in humid western Kenya (Millar et al., 2004). Compared to the $N_2O$ emissions reported for humid tropical maize production systems, our data suggest that maize-legume intercropping based on mulching in the sub-humid to semi-arid Rift valley appears to be a minor $N_2O$ source. Growing season $N_2O$

emission factors (EF) in our study ranged from 0.02 to 0.25 and 0.11 to 0.20% of the estimated total N input in 2015 and 2016, respectively, including assumed N inputs from legume mulch as well as belowground additions and carryover between the years (Table 1). Even if the estimated EF is doubled to account for off-season emissions, it is still lower than the annual IPCC default value of 1% $N_2O$-N per unit added N (IPCC, 2014). Our estimated EFs thus seem to be at the lower

end of those reported by Kim et al. (2016) for SSA smallholder agriculture estimated from literature data (0.01 to 4.1%). The reasons for the low EFs in our study are probably the high background emissions in the fertile soil of the Hawassa University research farm which supports high maize yields even in the unfertilized control (Table 1) and the low levels of N input. The soil has been used over decades for agronomic trials with various fertilization rates with and without

crop residue retention and legume intercropping (Raji et al., 2019). Thus, our field trial has to be considered representative for intensive management as opposed to smallholder systems with minimal or no fertilization history.



Methane uptake by the soil in both seasons varied between 1.0 to 1.5 kg $CH_4$-C $ha^{-1}$ without showing any significant treatment effect, even though maize-legume intercrops tended to take up

less $CH_4$ than maize monocrops (Fig. S1). The observed trend might relate to competitive inhibition of $CH_4$ oxidation by higher $NH_4^+$ availability (Le Mer and Roger, 2001, Dunfield and Knowles, 1995) in the presence of legume intercrops, even though estimated total N inputs remained below 100 kg N $ha^{-1}$, which is considered a threshold for $NH_4^+$ inhibition (Aronson and Helliker, 2010). Alternatively, densely growing legumes may have lowered $CH_4$ uptake through

impeding $CH_4$ and/or $O_2$ diffusion into the soil (Ball et al., 1997). We did not observe stimulation of $CH_4$ uptake by legume intercropping, which we attribute to the absence of N and P deficiency in this fertile soil. Methane uptake rates varied from 20 to 140 µg $CH_4$-C $m^{-2}$ $h^{-1}$ which is in the range of rates reported previously for SSA upland soils (Pelster et al., 2017). Seasonal $CH_4$ uptake in our experiment offset between 22 and 69% of the $N_2O$-GWP without revealing any significant

treatment effect (Fig. S1a and S1b), but the offset was relatively largest in the unfertilized maize monocrop and smallest in lablab intercropping. Hence, $CH_4$ uptake appears to be an important component of the non-$CO_2$ climate footprint of SSA crop production.

### 4.3 Legume intercropping and climate smart agriculture

Legumes are an important N source in smallholder farming systems, where mineral fertilizers are

unaffordable or unavailable. Legume intercrops maximize resource use efficiency as total productivity is often higher than in mono-cropping systems (Banik et al., 2006). Moreover, N fixed biologically by legume intercrops can partly replace synthetic N fertilizers, if the release is synchronized with the nutrient demand of the cereal crop. On the other hand, surplus N from legumes may result in N losses as $NO_3^-$, $NH_3$ and NO, $N_2O$ or $N_2$. Mulching and incorporation of

legume biomass has been found to increase $N_2O$ emissions under temperate conditions (Baggs et al., 2000, Baggs et al., 2003) and under humid tropical conditions (Millar et al., 2004). Also under semi-arid, Mediterranean conditions, vetch (*V. villosa*) used as a winter catch crop and mulched in spring significantly increased $N_2O$ emissions during the fallow period while rape did not (Sanz-Cobena et al., 2014). This was later confirmed by a [15]N study, highlighting the role of N

mineralization from legumes as a source of $N_2O$ (Guardia et al., 2016). None of the studies found an overall $N_2O$ saving effect of catch crops when scaling up to the entire crop cycle, even though the latter study used reduced mineral N fertilization rates in treatments with catch crops. By





contrast, reduced $NO_3^-$ leaching and $N_2O$ emission has been reported from maize intercropped with legumes in the semi-arid North China plain, which the authors attributed to enhanced N uptake by both the inter and main crop and reduced soil moisture in treatments with intercrops during the rainy season (Huang et al., 2017). This shows that legume intercrops have a potential to both increase or reduce $N_2O$ emissions with consequences for the non-$CO_2$ footprint of cereal production and hence for the viability of intercropping as a central component of CSA (Thierfelder et al., 2017).

The legume intercrops used in our study have low C:N ratios (Table S1), and can be expected to release a significant part of their N through decomposition of roots and nodules or root exudation as well as during decomposition of mulches (Fustec et al., 2010). The effect of mulching on $N_2O$ emissions depends on the C:N ratio of the residues with increased emissions for low C:N ratio residues (Baggs et al., 2000, Shan and Yan, 2013). In line with this, $N_2O$ emissions in intercrop treatments of our study exceeded those in fertilized maize monocrop on several sampling dates, both during active growth of legumes and after mulching. Another important aspect is the amount of legume N carried over between years which depends, among others, on amount and quality of the legume and the weather between the growing seasons. Abera et al. (2014) showed that surface-placed residues of haricot bean and pigeon pea decompose quickly despite relatively dry conditions during offseason. Vigorous rainfalls in the beginning of the growing season like in 2016 could lead to dissolved N losses, which will lead to indirect $N_2O$ emissions elsewhere or to elevated direct $N_2O$ emissions as seen on the first sampling date in 2016.

## 5. Conclusion

While legume intercrops have the potential to improve cereal yields and diversify produces for smallholders in SSA, a risk of enhanced $N_2O$ emissions remains, which became apparent as increased "$N_2O$ intensity" of the main crop in a drought year (2015). At the same time, our study points at possibilities to manage this risk by actively controlling legume biomass development and hence potential N input through "climate-smart" choices of legume species, sowing date and mulch amounts. Our study was conducted on a nutrient-rich soil which supports high yields of both maize and leguminous intercrops. Under these conditions, intercropped legumes can replace a considerable part of synthetic fertilizer, thus supporting common CSA goals. However, more



studies are needed to fully explore intercropping options in the framework of CSA in the East-African Rift Valley, particularly in nutrient-poor smallholder fields. Future studies on CSA approaches in SSA should address, in addition to non-$CO_2$ greenhouse gas emissions, N-runoff and soil organic matter build up, ideally in long-term field trials with and without legume intercropping. Given that seasonal $N_2O$ emission factors and intensities in our study were in the lower range of published values for SSA, intercropping appears as a promising approach to sustainable intensification in the Ethiopian Rift Valley.

**Acknowledgements.** The study is part of the NORHED program "Research and capacity building in climate smart agriculture in the Horn of Africa" funded by the Norwegian Agency for Development Cooperation (Norad). We are grateful to Teshome Geletu, Teketel Chiro and Tigist Yimer for assistance during setting up and managing the field experiment, sample collection and preparation.



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





Table 1: N inputs from forage legumes and fertilization per treatment which was estimated as outlined in the Materials and Method section 3.4. Shown are mean values (n=4 ± standard error)

| Legume | DMY | Aboveground N yield[a] | Belowground N yield[b] | N from mulch[c] | Mineral N | Carryover[d] | Total N input |
|---|---|---|---|---|---|---|---|
| | | | kg N ha[-1] | | | | |
| | | | 2015 | | | | |
| *Crotalaria* | | | | | | | |
| 3w | 1516±183 | 53.3±6.4 | 17.7±2.1 | 26.6±3.2 | 64 | | 75.8 |
| 6w | 345±65 | 12.1±2.3 | 4.0±0.8 | 6.1±1.1 | 64 | | 66.4 |
| | | | *Lablab* | | | | |
| 3w | 2221±340 | 96.8±14.8 | 32.3±4.9 | 48.4±7.4 | 64 | | 82.9 |
| 6w | 467±137 | 20.3±6.0 | 6.8±2.0 | 10.2±3.0 | 64 | | 67.7 |
| | | | 2016 | | | | |
| | | | *Crotalaria* | | | | |
| 3w | 468±85 | 16.4±3.0 | 5.47±1.0 | 8.21±1.5 | 32 | 11.1±1.3 | 56.8 |
| 6w | 65±44 | 2.3±1.5 | 0.75±0.5 | 1.13±0.8 | 32 | 2.5±0.5 | 36.4 |
| | | | *Lablab* | | | | |
| 3w | 1256±221 | 54.7±9.6 | 18.25±3.2 | 27.4±4.8 | 32 | 20.2±3.1 | 97.8 |
| 6w | 186±60 | 8.1±2.6 | 2.70±0.9 | 4.06±1.3 | 32 | 4.2±1.2 | 43.0 |

[a] N content of crotalaria and lablab was 3.51 and 4.36%, respectively, measured in 2 representative samples
[b] assuming a shoot-to-root ratio of 2 and an average belowground N input from the standing legumes of 50% during the growing season
[c] returning half of the aboveground yield as mulch; assuming an average N release of 50% and 30% for 3-week and 6-week treatments, respectively, during the growing season
[d] assuming that 50% of the remaining N becomes available in the following cropping season






Table 2: Grain yield, growing-season $N_2O$ emission factors and emission intensities for 107 and 123 days in 2015 and 2016, respectively and combined global warming potential (GWP) of $N_2O$ emission and $CH_4$ uptake for fertilized treatments with and without legume intercropping. N input was estimated as outlined 690 in Table 1. Shown are mean values (n=4 ± standard error). Different letters indicate statistical difference at $p < 0.05$.

| Treatment | 2015 | | | | 2016 | | | |
|---|---|---|---|---|---|---|---|---|
| | Maize Grain yield (kg ha$^{-1}$) | $N_2O$ emission factor (%) | *GWP (kg $CO_2$ eq. ha$^{-1}$ 107d$^{-1}$) | $N_2O$ emission intensity (g $N_2O$-N ton grain$^{-1}$) | Maize Grain yield (kg ha$^{-1}$) | $N_2O$ Emission factor (%) | *GWP (kg $CO_2$ eq. ha$^{-1}$ 123d$^{-1}$) | $N_2O$ emission intensity (g $N_2O$-N ton grain$^{-1}$) |
| Maize-F | 4313±235[a] | | 17.4±12[a] | 29.7±4.2[ab] | 6558±217[a] | | 29.7±18[a] | 26.3±4.0[a] |
| Maize+F | 5022±133[ab] | 0.07±0.07[ab] | 38.4±25[a] | 34.4±8.8[ab] | 8403±342[b] | 0.20±0.03[a] | 91.4±16[ab] | 37.0±4.0[a] |
| Maize+Cr3w | 5882±249[ab] | 0.17±0.05[ab] | 78.0±12[ab] | 42.2±5.5[b] | 8276±236[b] | 0.16±0.08[a] | 78.3±19[ab] | 33.6±4.7[a] |
| Maize+Cr6w | 5316±316[ab] | 0.07±0.06[ab] | 47.0±15[ab] | 34.8±5.4[ab] | 8283±148[b] | 0.16±0.05[a] | 69.0±12[ab] | 27.8±2.0[a] |
| Maize+Lb3w | 5989±528[b] | 0.25±0.06[b] | 120.5±27[b] | 54.3±6.1[ab] | 8557±262[b] | 0.15±0.03[a] | 111.7±9[b] | 36.8±2.1[a] |
| Maize+Lb6w | 5541±492[ab] | 0.02±0.01[a] | 21.2±7[a] | 24.6±1.5[a] | 8306±501[b] | 0.11±0.07[a] | 62.3±25[ab] | 26.8±3.9[a] |

* $N_2O$: 300 $CO_2$ eq; $CH_4$: 25 $CO_2$ eq



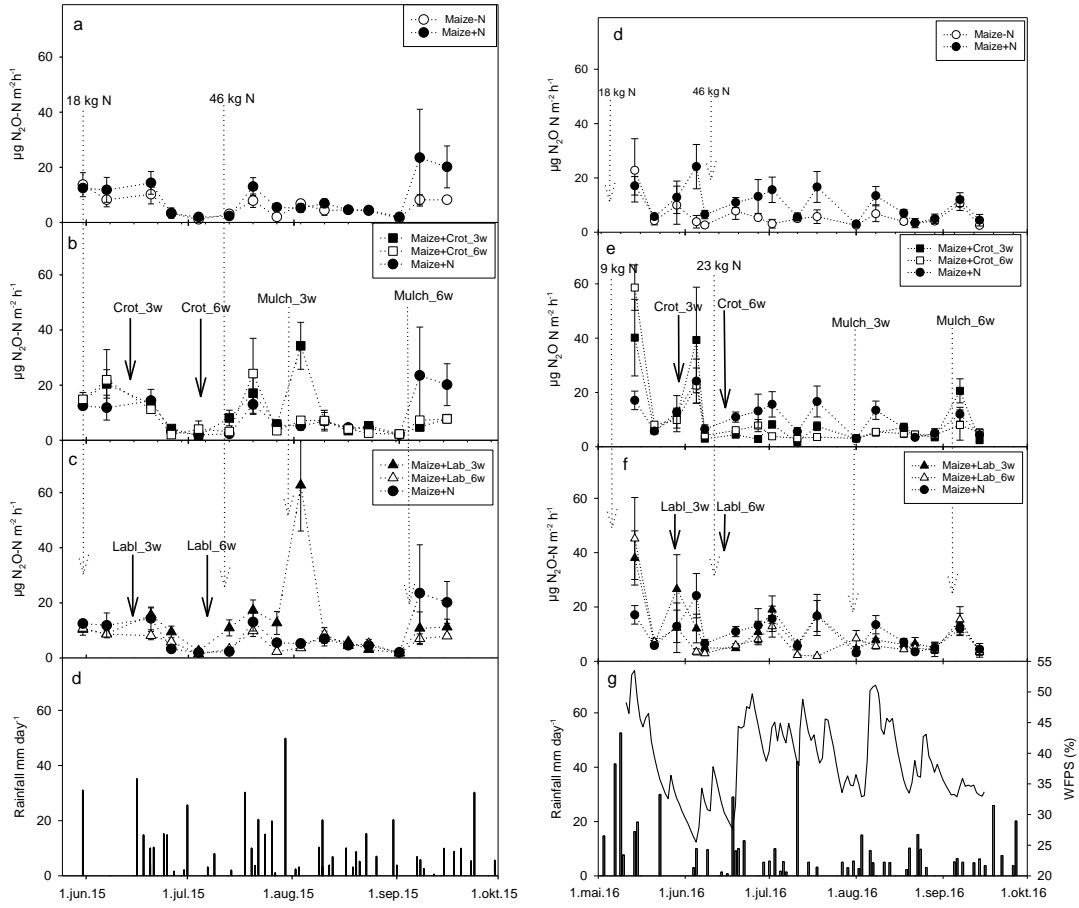

Figure 1: Mean N$_2$O emission rates (n=4; error bars = SEM) in 2015 (left panel) and 2016 (right panel) and daily rain fall and water-filled pore space (in 2016 only). Figures a and d show emission rates in the absence of intercrops, b and e with crotalaria and c and f with lablab intercrops.







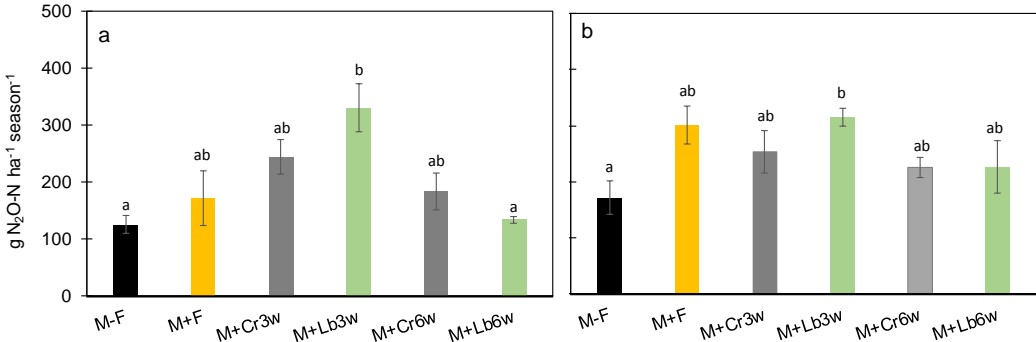

Figure 2: Cumulative seasonal $N_2O$-N (g N ha$^{-1}$ season$^{-1}$) in 2015 (a) and 2016 (b) throughout 107 and 123 days, respectively, in treatments with and without legume intercropping. Error bars denote

SEM (n=4). Different letters indicate significant differences at $p < 0.05$. M+F: fertilized maize; M+Cr3w: fertilized maize with crotalaria sown 3 weeks after maize; M+Cr6w: fertilized maize with crotalaria sown 6 weeks after maize; M+Lb3w: fertilized maize with lablab sown 3 weeks after maize; M+Lb6w: fertilized maize with lablab sown 6 weeks after maize




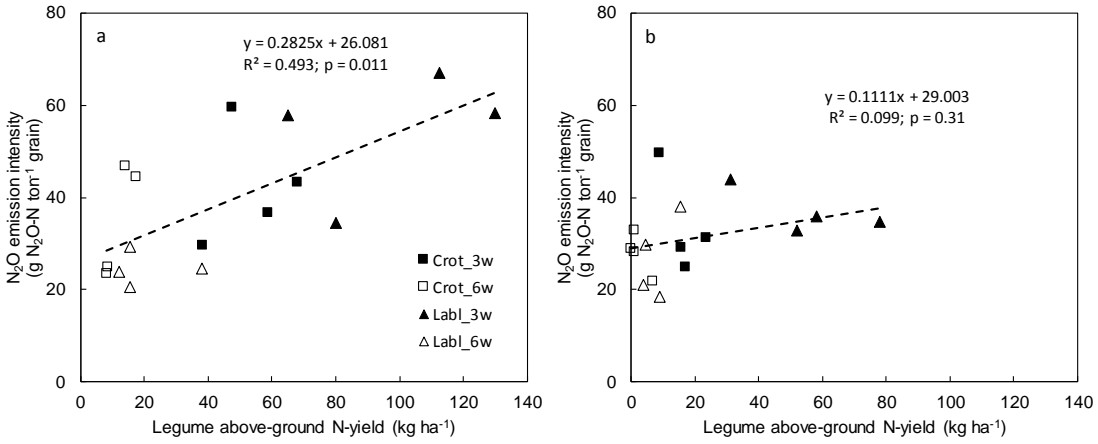


Figure 3: Relationship between $N_2O$ emission intensity and intercrop legume biomass yield in intercrop treatments in 2015 (a) and 2016 (b). Shown are single-plot values for each treatment (n=4).




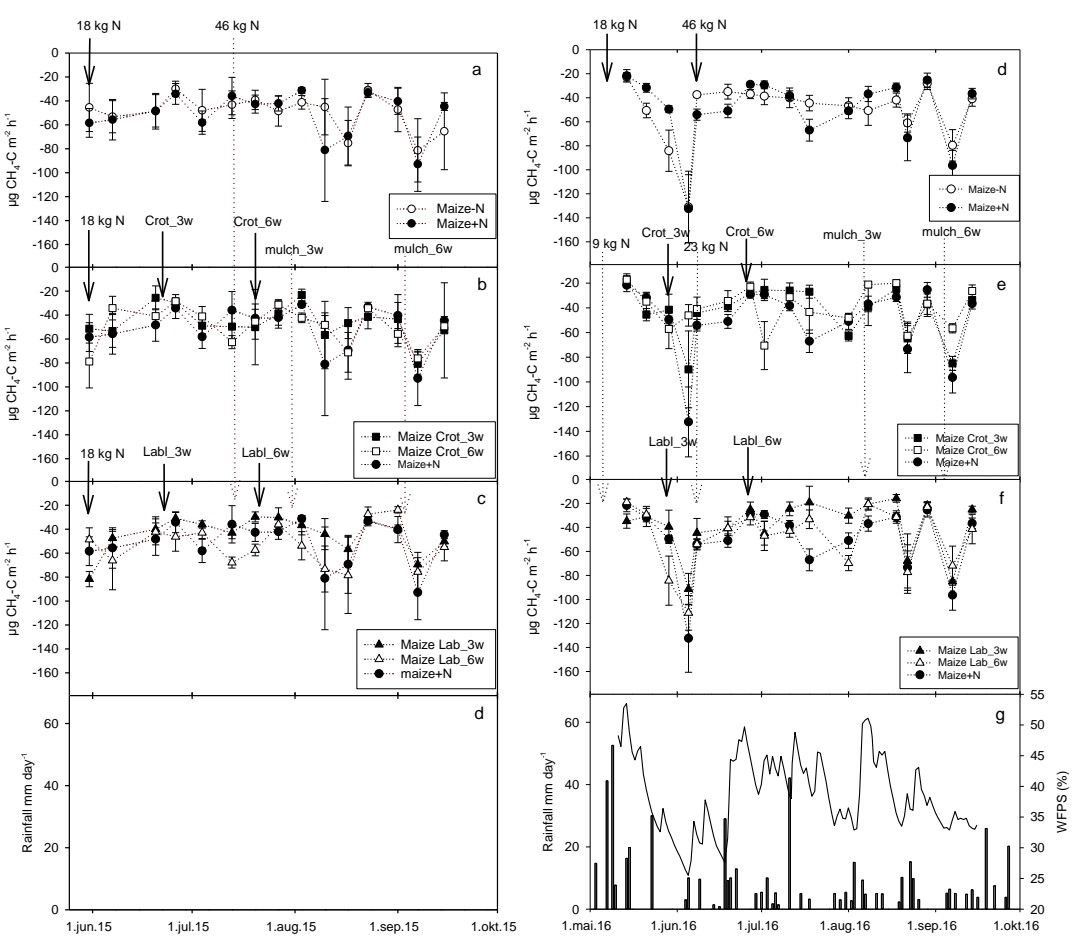

Figure 4: Mean CH$_4$ flux in 2015 (left panel) and 2016 (right panel) and daily rainfall and water-filled pore space (in 2016 only). Error bars show standard error of the mean (n=4). Figures a and d show emission rates in the absence of intercrops, b and e with crotalaria and c and f with lablab intercropping.







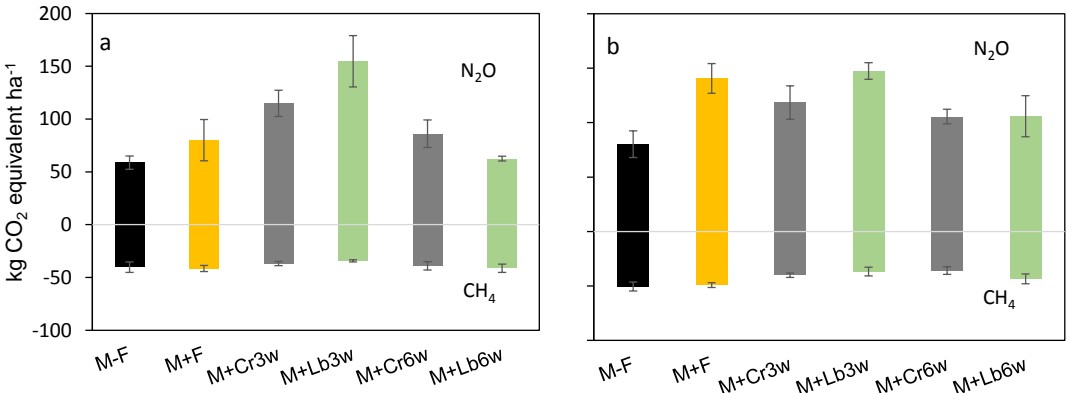

Figure 5: Relative contribution of CH$_4$ uptake and N$_2$O emission to seasonal GWP in mono- and intercropping treatments in 2015 (a) and 2016 (b). Error bars indicate standard deviation (n=4).

For treatment names, see Fig. 2.