# Peer review of "Effect of legume intercropping on N2O emission and CH4 uptake during maize production 1 2 in the Ethiopian Rift Valley 3 Shimelis G Raji1,2 and Peter Dörsch1 4 5 1 Faculty for Environmental Sciences and Resource Management, Norwegian University"

_Biogeosciences, 2019_

## Author Comment (AC1) · 19 Aug 2019

Due to a conversion error, figure 4d was lost. Please see the attached file for a correct version.

[Figure]

Figure 4: Mean $CH_4$ flux in 2015 (left panel) and 2016 (right panel) and daily rainfall and water-filled pore space (in 2016 only). Error bars show standard error of the mean (n=4). Figures a and d show emission rates in the absence of intercrops, b and e with crotalaria and c and f with lablab intercropping.

**Fig. 1.**

---

## Referee Comment (RC1) · Anonymous Referee #1 · 4 Sep 2019

General comments This paper reports results from an experiment conducted in Ethiopia measuring yields and GHG fluxes from maize cultivated as monocrop and intercropped with 2 legumes. There is an urgent need to increase the empirical base quantifying GHG fluxes from agricultural systems in Sub-Saharan Africa and therefore this study could be a valuable contribution to the literature. Understanding the interactions between cereal and legume crops and quantifying C footprints are also commendable scientific goals, and requirements to design future climate-smart farming. However, this study seems to have a number of experimental shortcomings that require at least clarification to be able to assess its suitability for publication in Biogeosciences. These are the most important issues to be addressed:

1. The introduction doesn't follow a logical flow. It includes interesting hypotheses, although the authors either do not properly attempt to answer the hypotheses or do it insufficiently. Example: "Legumes affect emissions by providing organic N or by modulating the completion between roots and microbes for soil N". The authors could have added how these processes are 'modulated', and use the appropriate methods to quantify species competition and microbial processes. 2. The methods are poorly described to assess the value of the experimental data. I indicated shortcomings in Specific comments below. 3. The discussion is mostly a compilation of literature conducted elsewhere reporting GHG fluxes from intercropping including legumes. I would expect a reflection of the results against the relevant literature.

A modest aim for this paper could have been simply documenting the GHG flux measurements and explaining the patterns observed, using all the data collected and conducting a sensitivity analysis for the fluxes that have been roughly estimated, such as the contribution of the legumes to N inputs, the emission factors and the emissions intensity.

Because there are very few experiments measuring GHG fluxes in Africa, I would suggest a thorough revision addressing the shortcomings, to re-consider this manuscript for publication.

Specific comments Introduction L39 The use of inorganic fertilisers doesn't necessarily reduce the soil methane sink. Please explain.

L40 remove 'by contrast'. It doesn't follow naturally from the previous sentence.

L41 the concept of CSA – coined by FAO – doesn't talk about profits. Please revisit original source.

L43 I don't think the understanding of GHG fluxes in SSA is limited. There is a scarcity of quantified GHG fluxes in SSA, and limited experimentation on which CSA practises would be suitable for the SSA context. Please rephrase.

L49 Crop production can be a major source of N2O emissions when fertilisers are used. This is not often the cause in East African agriculture. There are empirical studies that show that.

L53 strange reference to 'upland soils' here. Please explain why the focus is suddenly shifted towards upland soils.

L58 soil management practices are not the only controls of the factors affecting soil N2O fluxes. Soil type and climate are major determinants, which don't depend on management.

L59 The position of the two first references in this sentence is not logical. Please revise.

L68 diversification, rotation and intercropping do not always enhance productivity. Please rephrase.

L71 please add reference that shows that legume improve N uptake of the cereal crop in the Rift Valley (this is a large area across countries!). There is evidence in favour and against this.

L86 rates of 100 kg N per hectare are very uncommon in Africa. Please consult the literature on fertiliser use for the continent.

L89 increasing. Remove or replace 'accordingly', doesnt seem to fit the meaning of the sentece.

L93 add 'the' to 'the' release. Please explain how root exudates release 'extra N'.

L95-96: are these the hypotheses this study wanted to test?

L110-112: these hypotheses don't have any mechanistic underpinning, and are therefore weak. Time measured in weeks is unlikely to be a fixed effect, since the effects of management such as sowing date, choice of species and cultivar on yields and GHG fluxes will depend on soil and weather.

L115 because there are so few experiments measuring GHG fluxes in Africa, and more modest aim for this paper could have been simply documenting these measurements and explaining the patterns observed.

Methods L121-126 please report soil type using a known classification, e.g. WSD. And please add measure of dispersion to the reported soil properties, and weather variables.

L128 Please explain the 6 treatments clearly here. No clear which are the treatment is Table 1, and how they were imposed. Treatments seem to be listed in Table 2, although there is no consistency in labels used in Tables and Figures.

L130 only one cultivar? Wouldn't the reserachers have expected cultivar effects on the treatments?

L31 only one sowing date each year? I understood from the objective and hypothesis that the authors wanted to test the effect of sowing date (L110) on GHG fluxes.

L133 fertiliser rates per hectare? I am surprised to read that N fertiliser was applied to the intercropping treatment. Was there a scientific basis to half the rate? If yes, please add reference to previous experimental work.

L136 I would have expected an effect of plant density. These were fixed.

L141 why half removed? did you measure this variable amount of mulching applied to the plots? This is not really a welcome variation to the treatments, and could have affected the data analysis and assumptions on treatment effects.

L151 why didn't the measurements of fluxes start before planting to capture background GHG fluxes?

L152 what was the frequency of sampling? Weekly? There is evidence that less than weekly sampling doesn't capture the variation of GHG fluxes in a crop's cycle. See Barton et al. 2015 Scientific Reports volume 5, Article number: 15912 (2015)

L159 Helium filled?

L185 these treatments were not introduced before.

L187 Was bulk density measured? If yes, how?

L195-L198 Not having assessed belowground biomass and the amount of N fixed by the legumes is an important shortcoming of this study. Specially because the authors pose the hypothesis in the introduction (L95-96) that "Legumes affect emissions by providing organic N or by modulating the completion between roots and microbes for soil N". Without having quantified belowground N and N2 fixation, there results are less useful as a contribution to test this hypothesis.

L199 until here, it wasn't indicated that there were different sowing times for maize and legumes. Treatments must be clearly explained at the beginning.

L202-204 this is another shortcoming, having assumed the 'release' of 50% and 30% of the N during the growing season doesn't help with hypothesis testing. The authors could have followed at least inorganic N in the soil.

L213 this emission factor is not really meaningful given all the assumptions used to estimate N input.

L221 Was grain moisture content measured?

Results L236-237 to be able to measure peaks, N2O fluxes must be measured continuously after fertiliser application. There is typically a peak 6-48 hours after application. The dataset unfortunately doesn't show baseline emissions that happened before the treatments were imposed.

L 280-295 I find this section on EFs speculative because there are large uncertainties in the estimation of N input as described in the methods section.

L318 this should be explained in the methods section with all assumptions and reported as absolute emissions not GWP. This section is not clear, and need to consistently

explain Fig 2 and 5. Fig 5 doesn't include letters showing the contrasts.

L330-340 this belongs more to results than to discussion.

L349-354 because the researchers didn't measure N2 fixation, this sentence is speculative. Also attributing the lack of relationship between N input and legume N yield and N2O fluxes to the variability of fluxes is speculative, since the estimation of the N input and yield are very uncertain and based on strong assumptions.

L375-378 the data shown in Fig 2 doesn't show that intercropping legumes increases emissions 'risk' further than cultivating fertilised maize. If that were the case, there would be a consistent effect across years, and all legumes would increase emissions.

L381 unfortunately the experimental data of the one experiment in Ethiopia presented here is insufficient to claim that N2O fluxes in the sub-sequent year are negligible under SSA conditions. It is unfortunate that the researchers didn't follow the dynamics of inorganic N in the soil or plant N uptake when they sampled GHG fluxes.

L385 it is also unfortunate that the researchers don't present data of N2O fluxes and soil N dynamics off-season. So this observation remains speculative.

L385 not clear what is meant with 'emissions were at par', neither why this is striking.

L395 the lack of explanation to the effect on mulching actually calls to explain this by measuring consistently the factors driving N2O fluxes such as moisture content and availability of substrate (inorganic N) over time.

L397 the relative effect of soil moisture vs inorganic N could have been tested if the researchers would have collected such data. Now this conclusion leads to speculation.

L398-410 this study doesn't present solid evidence to sustain this claim, because sowing date doesn't control per se GHG fluxes, but determines the state of soil and weather that the soil+crop system will experience. So giving prescriptions of sowing dates that are not tied to indications of environmental conditions wouldn't be useful at all. In

addition, this research didn't find any consistent evidence that legumes increase the emissions beyond the fertilised crop according to Fig 2, which shows that one treatment had higher N2O fluxes than the control.

L412-420, this section needs re-writing to make a comparison instead of a list of studies and their findings.

L420-424 for this comparison to be useful, please report the biomass measured that was added in year 2 across treatments.

L428, in my opinion the EFs should be re-worked with uncertain parameter ranges to be able to assess how far there are from IPCC. This statement is too crude given the procedures used to estimate the EF.

L433 the levels of N inputs could have been underestimated because there were no measurements of the real contributions of the legumes. Which soil has been used over decades? Not clear. Intense use of soils usually leads to loss of fertility not enrichment.

L441 dynamics of inorganic N not measured.

L454-474 this piece of text is not needed because it cannot be compared with the experimental results reported here. I would suggest contrasting the experimental results with the literature and avoiding listing all that is known for legumes in completely different climates.

L482-482 I understood that the researchers didn't measure the N 'carry over effects'. So this point is speculative.

L485-487 this statement could be verified at least against the soil moisture data.

L494 please consider environmental conditions instead of referring to sowing date alone. You could also discuss what would be the incentives for farmers to reduce N2O emissions.

L500 indeed more studies would be needed to confirm and to explain the results obtained. I would suggest reflecting on the need to quantify N2 fixation, and to follow N mineralisation, especially key for legumes.

---

## Referee Comment (RC2) · Anonymous Referee #2 · 13 Sep 2019

This study looking at soil N2O and CH4 in agricultural systems of Sub-Saharan Africa addresses a significant gap in the body of literature exploring GHG exchange in intensively-managed soils, both through its location in an understudied area, and the aim to understand the relationship between inter-crop timing and N2O emissions. Although the article does need to be further edited for grammar/phrasing, it is generally well written. However, there are some issues with clarity I'd like to see addressed, which I expand on below.

Specific comments:

Note: Phrases in quotations are suggested changes.

Introduction

Line 40: Specifically define what CSA means in terms of management. The previous sentence defined intensification as 'increased use of inorganic fertilizers', and then CSA is introduced as, 'in contrast...' but the text doesn't in fact provide a contrast, instead outlining the ideals of the CSA concept.

Line 82: As you go on to explain, abundant NH4 can inhibit methanotrophs, but may not always. Important to make that distinction here.

**Materials and Methods**

In general, please try to provide as much detail as possible, grouping information in a way that it is easy to find.

Line 120: "The field experiment was conducted for two years (2015-2016) at the Hawassa..."

Line 128-145: List exactly what the six treatments were, before going on to give details about planting and fertilizer application. Also, be specific about what happened when in each treatment, including when and how the legumes were mulched and applied.

Line 147: Were there live plants in the chambers during sampling or were those first removed?

Line 149: Are the chambers used in this study the same as those in Rochette et al.? If not, as the chambers were custom-made, a bit more detail about them would be useful. Some information to include: The chambers did not have permanent bases, correct? How deep into the soil were they pressed? Was the volume provided in the text (Line 148), the volume before or after the chamber was pressed into the soil? How much time was there between deployment and the first sample? Were they always measured in the same location? Do you think that soil disturbance from deployment may have affected the samples? Were the chambers vented?
Line 153: The four samples were at 0, 15, 30 and 45 minutes? Or 15, 30, 45 and 60?

Line 172: Were all results less than R2=0.85 rejected? (I.e. were net 0 emissions/uptake rejected?) If so, do you think that may have biased your results?

**Results**

Line 243: "Irrespective of legume species, the highest emission rates..."

Line 244-247: What about the sixth treatment? Was it significantly different than that?

**Discussion**

In this section, it would be helpful to go back to the original hypotheses and specifically outline how the results compared and why.

Line 333: Provide range from Pelster et al.

Line 341-342: Is that consistent with other mulching studies?

Line 344: You provide a topic sentence here, which ends with: species, inter-cropping time and weather. I'd suggest following that up by expanding on each of those in the order you present them in that sentence.

Line 353: Can you provide a reference for 'notoriously high'?

Line 363-366: Remove details of how the data was analyzed (that is in the results section) and just focus on the meaning of the results shown in the figure.

Line 380-382: Is that consistent with other mulching studies?

Line 386-389: I don't understand this. Something was at par and then not significantly different? Please rephrase and perhaps provide a reference to the Table/Figure with the results that you are discussing.

Line 487: Provide reference to Table/Figure.

**Tables and Figures**
Note that these should always be able to stand alone (i.e. all necessary information required to understand them should be included). For all tables and figures, please define any abbreviations (i.e. Table 1 - DMY), remove references to previous sections (i.e. Table 1 - refer to M/M, Fig. 5 - refer to Fig. 2), and include basic information about the study (e.g. Table 1 - N inputs from forage legumes and fertilizer application in plots of maize inter-cropped with legumes 3 and 6 weeks after planting.)

Technical corrections:

Line 114/115: Rephrase.

Line 212: Capitalization.

Line 314: Remove neither/nor and just use 'or'.

There are many small editing errors in the Discussion that need to be corrected.

Some examples:

Line 334: Owing?

Line 337: "was too small"

Line 371: "owing to early"

Line 374: "legume and main crops"

Line 380: Capitalization

Table 1 – consider reformatting using spacing rather than lines, as the bold lines make it difficult to read

BGD

---

## Author Comment (AC2) · 26 Sep 2019

General comments: This paper reports results from an experiment conducted in Ethiopia measuring yields and GHG fluxes from maize cultivated as monocrop and intercropped with 2 legumes. There is an urgent need to increase the empirical base quantifying GHG fluxes from agricultural systems in Sub-Saharan Africa and therefore this study could be a valuable contribution to the literature. Understanding the interactions between cereal and legume crops and quantifying C footprints are also commendable scientific goals, and requirements to design future climate-smart farming. However, this study seems to have a number of experimental shortcomings that

require at least clarification to be able to assess its suitability for publication in Biogeosciences.

Response: We thank the reviewer for constructive comments and criticism. The reviewer's main points of critique can be summarized as i) lack of ancillary data (e.g. soil mineral N) and N-fluxes (e.g. quantification of BNF) and ii) too much speculation about underlying processes. We agree with the reviewer that our study has experimental shortcomings, but we believe that our research has some salient points worth communicating to a broader audience: i) Intercropping and mulching legumes to maize under Rift Valley conditions did not cause major N2O emissions, nor inhibit CH4 uptake during a dry and a wet year ii) Legume intercropping therefore appears as a viable option for climate-smart intensification which is urgently needed in the region iii) Even though being highly insecure, numbers of leguminous N input, N2O-EFs, etc. presented in our paper can be used as first estimates in the absence of better data We understand the reviewer's frustration about the lack of ancillary data (soil moisture in 2015, mineral N content, below ground legume biomass, etc.) but as in any empirical study, there are limitations to the number of variables which can be measured, particularly so when relying on local research infrastructure.

1. The introduction doesn't follow a logical flow. It includes interesting hypotheses, although the authors either do not properly attempt to answer the hypotheses or do it insufficiently. Example: "Legumes affect emissions by providing organic N or by modulating the completion between roots and microbes for soil N". The authors could have added how these processes are 'modulated', and use the appropriate methods to quantify species competition and microbial processes.

Response: Studying legume-rhizobia interactions is not trivial (see for example Raji et al., 2019). Species competition and microbial processes were not the primary focus of our study. Instead, we were interested in the overall effect of forage legume intercropping and its management on N2O and CH4 fluxes. We rephrased the sentence to ". . .or by modulating the competition between plants and microbes for soil N, for example by

acting as an additional N sink prior to nodulation".

2. The methods are poorly described to assess the value of the experimental data. I indicated shortcomings in Specific comments below.

Response: We address these shortcomings in response to the specific comments below.

3. The discussion is mostly a compilation of literature conducted elsewhere reporting GHG fluxes from intercropping including legumes. I would expect a reflection of the results against the relevant literature.

Response: Comparing our flux estimates with those found in other GHG studies in Sub-Saharan Africa is an important first step to scrutinize and contextualize our measurements. The remainder of the discussion tries to interpret treatment effects by linking fluxes to measured variables (weather, legume biomass, etc.), necessarily drawing on the general literature. We are not entirely sure what the reviewer means by "relevant literature". Even though intercropping with forage legumes is a common practice in the Ethiopian Rift Valley, there are no published studies on how these practices affect N2O and CH4 fluxes. We therefore compared our N2O fluxes and emission factors to those reported for humid tropical maize production systems with intercropping, which – at least geographically – come closest to the system studied by us. We would be grateful to learn about relevant literature we have missed out.

A modest aim for this paper could have been simply documenting the GHG flux measurements and explaining the patterns observed, using all the data collected and conducting a sensitivity analysis for the fluxes that have been roughly estimated, such as the contribution of the legumes to N inputs, the emission factors and the emissions intensity.

Response: Estimating emission factors necessitates estimating N inputs, which is particularly challenging in experiments involving N input from BNF, green manuring or

crop residue retention. Our estimates of N input by BNF are based on assumptions of legume shot-root ratios and residue decomposition rates, which we anchored in the literature, as outlined in chapter 2.4. We believe that this approach does not lend itself to "sensitivity analysis for the fluxes" as we do not use statistical models to explain variations in flux. We decided to abstain from such models because of the inherent insecurity of underlying variables such as legume N input. Instead, we resorted to simple linear regression using measured aboveground legume N yield and N2O emission intensity (Fig. 3). We want to emphasize that all variables and their derivations (cumulative flux, emission intensity and factors) were estimated or calculated on a per plot level before averaging them, giving at least some measure of dispersion (e.g. Figures 2, 5 and Table 2).

Because there are very few experiments measuring GHG fluxes in Africa, I would suggest a thorough revision addressing the shortcomings, to re-consider this manuscript for publication.

Response: A thoroughly revised manuscript will be provided addressing all points raised by the reviewers.

These are the most important issues to be addressed:

Specific comments Introduction

L39: The use of inorganic fertilisers does not necessarily reduce the soil methane sink. Please explain.

Response: No, it does not. Our introduction tries to detail the conditions potentially leading to reduced CH4 uptake by citing a meta-study that found an overall higher propensity for reduction in CH4 uptake at mineral N fertilization rates > 100kg N ha-1 y-1 (Aronson and Helliker, 2010). We further outline possible mechanisms regulating CH4 uptake in fields with intercropping in Lines 81-91 of the original text. At no point, we claim that inorganic fertilizers invariably reduce the soil's sink strength for methane.

[Figure]

The sentence in line 39 now reads: "Abundant ammonium ($NH_4^+$) may also reduce the soil $CH_4$ sink by competing with $CH_4$ for the active binding site of methane monooxygenase, the key enzyme of $CH_4$ oxidation (Bédard and Knowles, 1989)"

L40 remove 'by contrast'. It doesn't follow naturally from the previous sentence

Removed

L41 the concept of CSA – coined by FAO – doesn't talk about profits. Please revisit original source

Response: The reviewer is right. We remove 'profits'.

L43: I don't think the understanding of GHG fluxes in SSA is limited. There is a scarcity of quantified GHG fluxes in SSA, and limited experimentation on which CSA practices would be suitable for the SSA context. Please rephrase.

Response: We agree with the reviewer that sources and sinks of GHGs in SSA are well understood, in principal, and rephrase the sentence to "However, greenhouse gas emission measurements in SSA crop production systems are scarce and proof-of-concept for the mitigation potential of specific CSA practices is missing (Kim et al., 2016, Hickman et al., 2014b)."

L49: Crop production can be a major source of $N_2O$ emissions when fertilisers are used. This is not often the cause in East African agriculture. There are empirical studies that show that

Response: Food production in SSA has to double by 2050 to feed a growing population. This requires intensification of crop production, be it by increasing nitrogen fertilization or by other approaches, such as legume intercropping. We therefore believe that studying and documenting intensification effects on $N_2O$ emissions are important in the wider context of GHG mitigation in the global agrifood system. We agree that $N_2O$ emissions in rainfed SSA crop production appear small per date, but given the enormous productivity increase needed, also crop production in SSA may become a

major source of N2O. In the revised version, we state explicitly "Emission rates of N2O reported for SSA crop production so far are low (Kim et al., 2016) owing to low fertilization rates, but may increase with increasing intensification."

L53: strange reference to 'upland soils' here. Please explain why the focus is suddenly shifted towards upland soils

Response: The term "upland" was removed as the statement refers to factors that control N2O production in soils in general.

L58: soil management practices are not the only controls of the factors affecting soil N2O fluxes. Soil type and climate are major determinants, which don't depend on management

Response: The reviewer is right! We rephrase the statement and add soil type and climate as important factors for N2O emissions, including two new references. The sentence added reads: "Other important factors are soil type (Davidson et al., 2000) and temperature (Schaufler et al., 2010)."

L59: The position of the two first references in this sentence is not logical. Please revise.

Revised

L68: diversification, rotation and intercropping do not always enhance productivity. Please rephrase

Response: The reviewer is right. We rephrased the sentence to: "Crop diversification by combining legumes with cereals, both in rotation and intercropping, enhances overall productivity and resource use efficiency, if managed properly (Ehrmann and Ritz, 2014)"

L71 please add reference that shows that legume improve N uptake of the cereal crop in the Rift Valley (this is a large area across countries!). There is evidence in favor and

against this.

Response: A reference was added (Sime and Aune, 2018), describing the general benefits of legume inclusion in farming systems in the region.

L86 rates of 100 kg N per hectare are very uncommon in Africa. Please consult the literature on fertilizer use for the continent.

Response: This statement refers to the general relationship between N rates and methane uptake as elucidated by the meta-study of Aronson and Helliker (2010) and not to common N fertilization rates in SSA.

L89: increasing. Remove or replace 'accordingly', doesn't seem to fit the meaning of the sentence.

Done

L93: add 'the' to 'the' release. Please explain how root exudates release 'extra N'

Done

L95-96: are these the hypotheses this study wanted to test?

Response: No. This sentence refers to the background of how legumes may cause extra N2O emissions. The hypotheses of the study are given in L109 -115 in the original text.

L110-112: these hypotheses don't have any mechanistic underpinning, and are therefore weak. Time measured in weeks is unlikely to be a fixed effect, since the effects of management such as sowing date, choice of species and cultivar on yields and GHG fluxes will depend on soil and weather.

Response: The competition for nutrients after under-sowing the intercrop, as well as the benefit of the main crop from N transfer depend in deed on a variety of factors, particularly those which control the initial growth of maize and hence its shading effect

on the legume. Our study is a good illustration for this: equal sowing dates produced vastly different legume aboveground biomasses in a dry and a wet year (Table 1). Yet, among all factors, the sowing date of the legume (relative to the main crop) is the one, which potentially could be controlled by the farmer, preferably in response to prevailing weather conditions. In response to the reviewer's righteous remark, we modify the respective conclusion in the discussion section to "Our study therefore points to optimizing the sowing date in response to expected emergence and growth of maize as a promising option to control growth of the intercrop and hence to deal with the risk of increased N2O emissions associated with high legume biomass".

L115: because there are so few experiments measuring GHG fluxes in Africa, and more modest aim for this paper could have been simply documenting these measurements and explaining the patterns observed.

Response: Our study served two aims, documenting fluxes and evaluating intercropping strategies with respect to GHG mitigation. We agree that a merely descriptive study of fluxes in different treatments would have been the least risky approach, but mitigation needs causation if it is to be widely adopted. Therefore, we chose to link seasonal emissions to stipulated legume N input and climatic variability, which we believe are the key drivers for N2O emissions in sub-Saharan intercropping systems. Methods

L121-126 please report soil type using a known classification, e.g. WSD. And please add measure of dispersion to the reported soil properties, and weather variables.

Response: We now include the soil type and SD for the bulk density. For analysis of soil texture and chemical composition, we used composite soil samples and hence cannot give measures of dispersion.

L128: Please explain the 6 treatments clearly here. No clear which are the treatment is Table1,and how they were imposed. Treatments seem to be listed inTable2, although there is no consistency in labels used in Tables and Figures.

Response: A treatment list is now included in the Materials and Method section; label inconsistencies in tables and figures are corrected.

L130: only one cultivar? Wouldn't the researchers have expected cultivar effects on the treatments?

Response: Farmers' preference was considered in choosing the maize cultivar for the trial. This cultivar is widely used and we focused on legume species and intercropping times rather than maize cultivar as a factor.

L131: only one sowing date each year? I understood from the objective and hypothesis that the authors wanted to test the effect of sowing date (L110) on GHG fluxes.

Response: Our objective was to test the effect of legume species and sowing date of the legumes relative to maize in combination with interannual weather variation, and not the effect of the sowing date of maize itself.

L133: fertiliser rates per hectare? I am surprised to read that N fertiliser was applied to the intercropping treatment. Was there a scientific basis to half the rate? If yes, please add reference to previous experimental work.

Response: Mineral N fertilization followed national recommendations, which are low. Annual legume intercrops are used, among others, to bring additional nitrogen into the soil, both during growth and after harvest. As outlined in line 134 ff., the rate of annual mineral N fertilization was halved in the second year there where legume mulch was applied, to test whether biologically fixed N could replace mineral N, which in itself would be a climate-smart approach. Cutting down on mineral fertilization is a common goal and practice when using catch or cover crops as green manure.

L136 I would have expected an effect of plant density. These were fixed.

Response: The numbers given for legume density are the planting densities, which did not result in "fixed" densities during the growing season. Much to the contrary, in terms of legume aboveground biomass, there was a huge variability across the two years.

Aboveground dry matter varied from 186 to 2221 kg ha-1 for lablab and 65 to 1516 kg ha-1 for crotalaria across the two years. We used this variation to explore the effect of legume biomass on N2O emissions, which indeed showed a significant effect in the dry year 2015 (Figure 3).

L141: why half removed? did you measure this variable amount of mulching applied to the plots? This is not really a welcome variation to the treatments, and could have affected the data analysis and assumptions on treatment effects.

Response: The idea of removing 50% of the biomass and mulching the rest was motivated by livestock feed shortage in the mixed farming systems of the region. Providing feed through intercropping provides an added "climate smart" value by alleviating the pressure on crop residues otherwise used as feed, thus increasing residue retention and building/stabilizing soil carbon. It is true that different mulching rates introduced additional variation. However, applying equal mulching rates to all plots would have negated plot-specific differences in soil fertility and hence belowground input. We therefore decided to scale the rate of mulch applied according to the plot-wise legume yield, as would be done by practitioners in real fields. In this way, we created a wide range of legume biomasses and likely also of N inputs, which allowed us to explore the effect of legume growth on N2O emissions (Fig. 3).

L151: why didn't the measurements of fluxes start before planting to capture background GHG fluxes?

Response: The flux study was restricted to two growing season due to logistic reasons. Two control treatments with maize monocrops were included, one with recommended mineral fertilization and one without. Thus, background fluxes are captured. We agree that flux measurements outside the cropping season would be desirable.

L152: what was the frequency of sampling? Weekly? There is evidence that less than weekly sampling doesn't capture the variation of GHG fluxes in a crop's cycle. See Barton et al. 2015 Scientific Reports volume 5, Article number: 15912 (2015)

Response: Flux sampling was conducted weekly as indicated in line 151 of the original manuscript

L159: Helium filled? Yes. Corrected.

L185: these treatments were not introduced before.

Response: The treatments M+Cr3w and M+Lb3w are now introduced at the beginning of the Materials and Method section.

L187 Was bulk density measured? If yes, how?

Response: A description of how it was measured is now added.

L195-L198: Not having assessed belowground biomass and the amount of N fixed by the legumes is an important shortcoming of this study. Especially because the authors pose the hypothesis in the introduction (L95-96) that "Legumes affect emissions by providing organic N or by modulating the completion between roots and microbes for soil N". Without having quantified belowground N and N2 fixation, the results are less useful as a contribution to test this hypothesis.

Response: The question of whether and how biologically fixed N affects N2O emissions is a long-standing issue. Our study is a modest attempt to address this issue for sub-Saharan conditions. It was however not designed to capture the exact mechanisms of competition between crops and microbes, nor did we hypothesize that it would. Instead, our working hypothesis was that legumes inter-cropped early in the season would increase N2O emissions if fertilized at the same time (L. 113). The reviewer is right that determining the amount and N content of belowground biomass would strengthen our approach, but given the number of field plots and the clayey soil (which makes it difficult to extract roots), the effort to do so would have been exorbitant. We therefore used aboveground biomass and its N content as a proxy for "potential" legume N input by scaling up literature based shoot-root ratios for lablab and crotalaria and estimating N release factors from literature.

L199: until here, it wasn't indicated that there were different sowing times for maize and legumes. Treatments must be clearly explained at the beginning.

Response: Additional explanations about the treatments have been added to the Materials and Methods section

L202-204: this is another shortcoming, having assumed the 'release' of 50% and 30% of the N during the growing season doesn't help with hypothesis testing. The authors could have followed at least inorganic N in the soil.

Response: We agree that mineral N contents would have supplemented our dataset in a meaningful way, but frozen storage of extracts prior to shipment out of the country was not possible due to frequent power cuts. As to the estimated release factors for legume N in the two years, we give detailed rationale for the underlying assumption (L.201 – 212, original version).

L213: this emission factor is not meaningful given all the assumptions used to estimate N input.

Response: We agree in principal, but seasonal emission factors for N2O have been used in the literature previously and may be considered useful for comparing different crop management strategies in regions with scarce flux data (see f. ex. Kim et al., 2016)

L221 Was grain moisture content measured?

Response: Yes, we used a digital grain moisture meter. Results

L236-237: to be able to measure peaks, N2O fluxes must be measured continuously after fertiliser application. There is typically a peak 6-48 hours after application. The dataset unfortunately doesn't show baseline emissions that happened before the treatments were imposed.

Response: The term "peaks" has been removed

[Figure]

L 280-295: I find this section on EFs speculative because there are large uncertainties in the estimation of N input as described in the methods section.

Response: See answer to L. 213 above.

L318: this should be explained in the methods section with all assumptions and reported as absolute emissions not GWP. This section is not clear, and need to consistently explain Fig 2 and 5. Fig 5 doesn't include letters showing the contrasts.

Response: We thank the reviewer for drawing our attention to missing indicators in Figure 5. GWP is replaced with "total non-CO2 GHG emissions" as suggested. Discussion

L330-340: this belongs more to results than to discussion.

Response: See our response to your general comment #3

L349-354: because the researchers didn't measure N2 fixation, this sentence is speculative. Also attributing the lack of relationship between N input and legume N yield and N2O fluxes to the variability of fluxes is speculative, since the estimation of the N input and yield are very uncertain and based on strong assumptions.

Response: Therefore, we talk about "potential leguminous N input" and not actual N input. We agree that our estimates of N input are insecure, but our analysis does not do more than examining the relationship of cumulative N2O emissions and "potential leguminous N input" on a plot for plot basis, before problematizing this approach in the discussion following L. 349.

L375-378: the data shown in Fig 2 doesn't show that intercropping legumes increases emissions 'risk' further than cultivating fertilized maize. If that were the case, there would be a consistent effect across years, and all legumes would increase emissions

Response: We believe that the sentence "Our data suggest that excessive accumulation of leguminous biomass in SSA maize cropping enhances the risk for elevated

N2O emissions" summarizes our findings in an appropriate way based on the analysis shown in Figure 3. In the discussion following L. 275, we are explicit about other factors such as rainfall early in the season potentially overriding this relationship.

L381: unfortunately the experimental data of the one experiment in Ethiopia presented here is insufficient to claim that N2O fluxes in the sub-sequent year are negligible under SSA conditions. It is unfortunate that the researchers didn't follow the dynamics of inorganic N in the soil or plant N uptake when they sampled GHG fluxes.

Response: We agree. Therefore, we added a disclaimer to this paragraph (L. 383-395). Future studies will examine N carry over between cropping seasons following mulching of the legumes in more detail, involving mineral N measurements and nutrient modelling.

L385: it is also unfortunate that the researchers don't present data of N2O fluxes and soil N dynamics off-season. So this observation remains speculative.

Response: Unfortunate, yes, but at least we draw N leaching into consideration.

L385: not clear what is meant with 'emissions were at par', neither why this is striking.

Response: Emission "at par" means emissions were at the same level. To avoid further confusion, we replace this expression with "comparable".

L395: the lack of explanation to the effect on mulching actually calls to explain this by measuring consistently the factors driving N2O fluxes such as moisture content and availability of substrate (inorganic N) over time.

Response: Absolutely! The sentence now reads: ". . . calls for studies tracing cumulative mulching effects over multiple years and exploring their driving factors in more detail."

L397: the relative effect of soil moisture vs inorganic N could have been tested if the researchers would have collected such data. Now this conclusion leads to speculation.

Response: Yes; see answer above.

L398-410: this study doesn't present solid evidence to sustain this claim, because sowing date doesn't control per se GHG fluxes, but determines the state of soil and weather that the soil+crop system will experience. So giving prescriptions of sowing dates that are not tied to indications of environmental conditions wouldn't be useful at all. In addition, this research didn't find any consistent evidence that legumes increase the emissions beyond the fertilized crop according to Fig 2, which shows that one treatment had higher N2O fluxes than the control.

Response: Sowing date of legumes in our study had a clear effect on legume development (aboveground biomass yield). We agree that our data provide no basis for prescribing sowing dates, and it was never our intention to do so. The sentence now reads: ". . . emerge as viable management factors for controlling the accumulation of legume biomass between the maize rows and hence the risk for increased N2O emissions". See also our response to L. 110.

L412-420,this section needs re-writing to make a comparison instead of a list of studies and their findings.

Response: We do compare emission factors; see line 423 in the original version

L420-424 for this comparison to be useful, please report the biomass measured that was added in year 2 across treatments.

Response: See Table 1

L428, in my opinion the EFs should be re-worked with uncertain parameter ranges to be able to assess how far there are from IPCC. This statement is too crude given the procedures used to estimate the EF.

Response: We agree but want to point out that we use our emission factors solely to compare our numbers with emission factors compiled by Kim et al. (2016) for other SSA agricultural systems, which also were scaled up from a limited number of measurements. It is by no means our intention to challenge IPCC default values. One may be critical to the concept of IPCC Tier 1 emission factors for regions with few flux data, but they are the only tool, for the time being, to compare systems with respect to their propensity to emit N2O from added reactive N and hence an important criterion when studying intensification effects.

L433 the levels of N inputs could have been underestimated because there were no measurements of the real contributions of the legumes. Which soil has been used over decades? Not clear. Intense use of soils usually leads to loss of fertility not enrichment.

Response: Smallholder farmer in the Rift Valley use little if any fertilization and remove all crop residues for feeding animals, thus they have a negative nutrient balance and lose soil fertility. In comparison, the experimental fields of the university farm are relatively "fertile" as they have experienced N and P fertilization, residue retention and N input from legumes BNF for many years. We believe it is important to point this out, when generalizing our findings for the region.

L441: dynamics of inorganic N not measured.

Response: Yes; that is why we have to speculate here.

L454-474: this piece of text is not needed because it cannot be compared with the experimental results reported here. I would suggest contrasting the experimental results with the literature and avoiding listing all that is known for legumes in completely different climates.

Response: This part discusses benefits and risks of legumes intercropping from the perspective of smallholder farming in the region. We believe that this discussion is important and integral to CSA and the question how to sustainably intensify crop production. Since there are no published studies on legume intercropping covering GHG emissions from this region so far, we resort to similar studies in other regions, trying to relate legume quality and management to nutrient release and N2O emissions.

[Figure]

L482-482: I understood that the researchers didn't measure the N 'carry over effects'. So this point is speculative.

Response: Yes, we speculate here.

L485-487: this statement could be verified at least against the soil moisture data.

Response: Daily rainfall is shown in figures 1d, 1g and 4d, 4g.

L494: please consider environmental conditions instead of referring to sowing date alone. You could also discuss what would be the incentives for farmers to reduce N2O emissions.

Response: We agree. We added a sentence reading: "This is complicated by the annual variability in growth conditions and requires active planning of sowing and mulching time by the farmers."

L500: indeed more studies would be needed to confirm and to explain the results obtained. I would suggest reflecting on the need to quantify N2 fixation, and to follow N mineralisation, especially key for legumes.

Response: We fully agree and have changed the sentence accordingly: "Future studies should attempt to combine flux measurements with inorganic N dynamics and BNF measurements".

---

## Author Comment (AC3) · 26 Sep 2019

This study looking at soil N2O and CH4 in agricultural systems of Sub-Saharan Africa addresses a significant gap in the body of literature exploring GHG exchange in intensively-managed soils, both through its location in an understudied area, and the aim to understand the relationship between inter-crop timing and N2O emissions. Although the article does need to be further edited for grammar/phrasing, it is generally well written. However, there are some issues with clarity I'd like to see addressed, which I expand on below.

Specific comments: Note: Phrases in quotations are suggested changes.

Response: We thank the reviewer for recognizing the validity of our study, and particularly for noticing our efforts to elucidate the relationship between intercrop timing, legume biomass development and N2O emissions.

Introduction Line 40: Specifically define what CSA means in terms of management. The previous sentence defined intensification as 'increased use of inorganic fertilizers', and then CSA is introduced as, 'in contrast...' but the text doesn't in fact provide a contrast, instead outlining the ideals of the CSA concept.

Response: The reviewer is right. We remove 'by contrast' because there is no contrast.

Line 82: As you go on to explain, abundant NH4 can inhibit methanotrophs, but may not always. Important to make that distinction here.

Response: We agree. The sentence has been rephrased using conditional "... might inhibit methanotrophs" to avoid misunderstanding.

Materials and Methods

In general, please try to provide as much detail as possible, grouping information in a way that it is easy to find.

Line 120: "The field experiment was conducted for two years (2015-2016) at the Hawassa..."

Rephrased

Line 128-145: List exactly what the six treatments were, before going on to give details about planting and fertilizer application. Also, be specific about what happened when in each treatment, including when and how the legumes were mulched and applied.

Response: Thank you for drawing our attention to this omission. We now added a detailed treatment description including the exact timing of mulching.

Line 147: Were there live plants in the chambers during sampling or were those first

removed? Response: Legumes were included in the chambers, on average 3 lablab plants and 4-5 crotalaria plants.

Line 149: Are the chambers used in this study the same as those in Rochette et al.? If not, as the chambers were custom-made, a bit more detail about them would be useful. Some information to include: The chambers did not have permanent bases, correct? How deep into the soil were they pressed?

Response: No, they were not identical to the chambers devised by Rochette et al. (2008). By accident we cited the wrong study by Rochette et al. (2008). This ref has now been replaced by Rochette, P., Eriksen-Hamel, N.S.: Chamber measurements of soil nitrous oxide flux: Are absolute values reliable? Soil Sci. Soc. Am. J., 72, 331-342, 2008, which gives a general outline of the static chamber method. The chambers did not have permanent bases but were pushed gently about 3 cm into the soil and sealed with moist clay from outside. The insertion depth is now added to the text.

Was the volume provided in the text (Line 148), the volume before or after the chamber was pressed into the soil? How much time was there between deployment and the first sample? Were they always measured in the same location? Do you think that soil disturbance from deployment may have affected the samples? Were the chambers vented?

Response: The number given in the text denotes the chamber volume after pushing it into the soil. The chambers were deployed randomly within the same maize row of each treatment plot to avoid disturbance. The chambers were not vented, but the sampling septum was removed when pressing the chambers into the soil to avoid perturbation of the concentration gradient. This information has now been added.

Line 153: The four samples were at 0, 15, 30 and 45 minutes? Or 15, 30, 45 and 60?

Response: Immediately after closing the chamber and sealing with soil, sampling starts (1 minute) and then at 15 minute intervals, hence 0, 15, 30, 45 minutes. The text has

been changed accordingly.

Line 172: Were all results less than R2=0.85 rejected? (I.e. were net 0 emissions/uptake rejected?) If so, do you think that may have biased your results?

Response: No fluxes were rejected. Regression coefficients were generally >0.85

Results

Line 243: "Irrespective of legume species, the highest emission rates..."

Corrected

Line 244-247: What about the sixth treatment? Was it significantly different than that?

Response: Thank you for drawing our attention to this. $N_2O$ emissions were significantly higher than in the fertilized control in both the 3-weel lablab and the 3-week crotalaria systems. The text is changed accordingly.

Discussion

In this section, it would be helpful to go back to the original hypotheses and specifically outline how the results compared and why.

Response: We added a sentence contrasting the findings discussed in chapter 4.1. with our original hypothesis.

Line 333: Provide range from Pelster et al.

Done

Line 341-342: Is that consistent with other mulching studies?

Response: There are not many studies to compare our results with, particularly not in SSA. Moreover, findings on the effect of mulching on $N_2O$ emissions are inconsistent, presumably because they depend on weather (soil moisture) as in our study. See also Basche et al. (2014), doi:10.2489/jswc.69.6.471

Line 344: You provide a topic sentence here, which ends with: species, inter-cropping time and weather. I'd suggest following that up by expanding on each of those in the order you present them in that sentence.

Response: The text is now rearranged and expanded following the reviewer's suggestion.

Line 353: Can you provide a reference for 'notoriously high'?

Response: We added Flessa et al (1995) who measured in various cropping systems, including cover and catch crops.

Line 363-366: Remove details of how the data was analyzed (that is in the results section) and just focus on the meaning of the results shown in the figure.

Removed

Line 380-382: Is that consistent with other mulching studies?

Response: Increased N cycling in spring after mulching is occasionally observed. We added Campiglia et al. (2011) as a reference for this

Line 386-389: I don't understand this. Something was at par and then not significantly different? Please rephrase and perhaps provide a reference to the Table/Figure with the results that you are discussing.

Rephrased

Line 487: Provide reference to Table/Figure.

Done

Tables and Figures

Note that these should always be able to stand alone (i.e. all necessary information required to understand them should be included). For all tables and figures, please define any abbreviations (i.e. Table 1 – DMY), remove references to previous sections (i.e. Table1–refer to M/M, Fig. 5–refer to Fig. 2), and include basic information about the study (e.g. Table 1 – N inputs from forage legumes and fertilizer application in plots of maize inter-cropped with legumes 3 and 6 weeks after planting.)

Response: We thank the reviewer for the these editorial remarks which we follow eagerly

Technical corrections:

Line 114/115: Rephrase.

Response: The sentence was rephrased to: "Choosing legume species, and sowing date and accounting for potential N inputs from legume intercrops, thus could allow to for better management of legume intercropping in SSA with reduced GHG emissions"

Line 212: Capitalization.

Done

Line 314: Remove neither/nor and just use 'or'.

Done

There are many small editing errors in the Discussion that need to be corrected. Some examples:

Line 334: Owing?

Rephrased

Line 337: "was too small"

Fixed

Line 371: "owing to early"

Fixed

Line 374: "legume and main crops"

Fixed

Line 380: Capitalization

Table 1 – consider reformatting using spacing rather than lines, as the bold lines make it difficult to read

Reformatted

---

## Author Response (AR2)

Response letter to bg-2019-303

We thank the editor and reviewer #2 for their renewed efforts improving our manuscript "Effect of legume intercropping on N2O emission and CH4 uptake during maize production in the Ethiopian Rift valley" and respond as follows:

Editor:

reviewer #2 has read the revised manuscript and made some comments and suggestions for some final changes. I agree with the suggestions especially think that you should make clear what is meant with CSA? If it is more than just a buzz-word that you can definitely be more specific in your manuscript on what 'practices' are meant. If it really is only a buzz-word, I think you should consider to remove it from your manuscript

Reviewer #2:

**Introduction**
Line 40: It still isn't clear what is meant by CSA. What does climate-smart agriculture actually mean? If it is a proposed way forward, what way is being proposed?

We now have added text to the introduction detailing what CSA is (in terms its components) and how it is connected to our study (L.40 in R2).

In Line 44 you mention 'proof-of concept for …specific CSA practices'. Which specific practices? If what is meant here is instead that we need practices that are 'climate-smart' but we haven't yet figured out what those practices might be, then this paragraph should be rephrased to make that clear. Otherwise, can you list 'climate-smart' practices that have been proposed but still need to be tested? I.e. are you referring to the use of legumes? If so, mention that specifically.

We agree with the reviewer that the formulation "specific CSA practices" is rather unspecific. We now list CSA practices and show how legume intercropping relates to CSA (L. 40 ff.) and replace "specific CSA practices" with "legume intercropping" (L. 53)

**Materials and Methods**
Line 134-135: change the order of this sentence so that it ends with "six treatments: [list of treatments…]" rather than "four replicates: [list of treatments…]"

-Rearranged

Line 139: avoid using respectively when possible. Here, just state: "on May 30, 2015 and May 7, 2016"
-Done

Line 141: state specifically that all treatments except the unfertilized control received fertiliser application; also consider using "MF+…" for your intercropping treatments to make that clear.
-Done

Line 149: avoid using respectively when possible. Here, just state: "July 27 and Sept 4, 2015 and Aug 2 and Sept 8, 2016"

Done

Line 157-160: Can you include a picture as supplementary information, so it is clear what the method looked like? Were the locations truly random (randomly generated) or just different each week? Also, consider rephrasing the last sentence to include more information about deployment. I.e. "…were pushed gently ~3 cm into the soil and included 2-5 live legume plants in the head-space. The septum was left open during deployment; once the chamber was inserted into the soil, the septum was closed and the base of the chamber was sealed around the circumference using moist clay."

A photograph of the chamber deployed between the rows has been added to the Supplementary Information.

Locations within each plot used for flux measurements were not generated randomly but chosen to cover small-scale variability within the middle rows and to avoid damaging legume roots.

The sentence describing chamber deployment has been revised according to the reviewer's suggestions.

Line 183: Specify here that zero fluxes were included and, if correct, that coefficients were generally (as opposed to always) greater than 0.85.

We indeed did not discard any fluxes because of low R2 values, which would heavily bias the data towards higher fluxes. This is now specified in L. 203.

General comment: as mentioned above, minimize the use of 'respectively'. It will make the text easier to read. Suggested changes are to Lines: 25, 256, 269-270, 297, 427, 442 and Table 2.

-Done

**Results**
Line 254: "generally larger" is only true for the 3-week fluxes, so perhaps specify that, or list the 3-week ones followed by the six-week ones.

-3-week fluxes listed first as suggested

Line 264: be consistent with treatment names; here the 'M-F' treatment is called 0N-control, elsewhere it is called the '0N0P treatment' or the 'unfertilized control' or the 'unfertilized maize monocrop'

-Fixed

Line 268: give units

-Done

[revised manuscript text omitted]